# A TRANSFER ATTACK TO IMAGE WATERMARKS

**Yuepeng Hu, Zhengyuan Jiang, Moyang Guo, Neil Zhenqiang Gong**
Duke University
`{yuepeng.hu, zhengyuan.jiang, moyang.guo, neil.gong}@duke.edu`

## ABSTRACT

Watermark has been widely deployed by industry to detect AI-generated images. The robustness of such watermark-based detector against evasion attacks in the white-box and black-box settings is well understood in the literature. However, the robustness in the *no-box* setting is much less understood. In this work, we propose a new transfer evasion attack to image watermark in the no-box setting. Our transfer attack adds a perturbation to a watermarked image to evade multiple surrogate watermarking models trained by the attacker itself, and the perturbed watermarked image also evades the target watermarking model. Our major contribution is to show that, both *theoretically* and *empirically*, watermark-based AI-generated image detector based on existing watermarking methods is not robust to evasion attacks even if the attacker does not have access to the watermarking model nor the detection API. Our code is available at: https://github.com/hifi-hyp/Watermark-Transfer-Attack.

## 1 INTRODUCTION

Generative AI (GenAI) can create highly realistic images, raising concerns about online information authenticity. Watermarking (Bi et al., 2007; Zhu et al., 2018; Tancik et al., 2020; Zhang et al., 2020; Al-Haj, 2007; Abdelnabi & Fritz, 2021; Kirchenbauer et al., 2023) was highlighted as a key technology for distinguishing AI-generated content in the White House's October 2023 Executive Order on AI security. In this approach, a watermark is embedded in AI-generated images, which can be decoded to verify the image's origin. Watermarking is widely used, with examples like Google's SynthID for Imagen (Saharia et al., 2022), OpenAI's watermark for DALL-E (Ramesh et al., 2021), and Stable Diffusion's user-enabled watermarking (Rombach, 2022).

An attacker can use *evasion attacks* (Jiang et al., 2023; Zhao et al., 2024; Saberi et al., 2024; Lukas et al., 2024) to remove a watermark from an image and evade detection. This involves adding a perturbation so that the watermark-based detector falsely identifies the image as non-AI-generated. The robustness of such detectors against evasion attacks has been studied in both *white-box* (the attacker has access to the watermarking model) and *black-box* (the attacker can query the detection API) settings (Jiang et al., 2023). In white-box attacks, small perturbations evade detection while preserving image quality, while in black-box attacks, perturbations are found by querying the detection API multiple times.

However, robustness of watermark-based detection in the *no-box setting* (the attacker even lacks access to the detection API) is less understood. In this setting, attackers may use *common post-processing* (e.g., JPEG compression) or *transfer attacks* (Jiang et al., 2023; An et al., 2024). Transfer attacks involve perturbing a watermarked image using surrogate models, which can be *classifier-based* or *watermark-based*. Classifier-based attacks treat the detector as a binary classifier and apply adversarial examples (Liu et al., 2017; Chen et al., 2024), while watermark-based attacks train a surrogate watermarking model for white-box attacks (Jiang et al., 2023). Both methods have limited success against advanced watermarking, leading to a misleading conclusion in prior studies that watermarking is robust in the no-box setting.

**Our work:** In this work, we propose a new watermark-based transfer attack in the no-box setting to evade AI-generated image detection. Unlike classifier-based attacks (An et al., 2024), our approach directly uses surrogate watermarking models, making it more suited to watermark-based detection. Unlike previous watermark-based attacks (Jiang et al., 2023), we employ multiple surrogate models

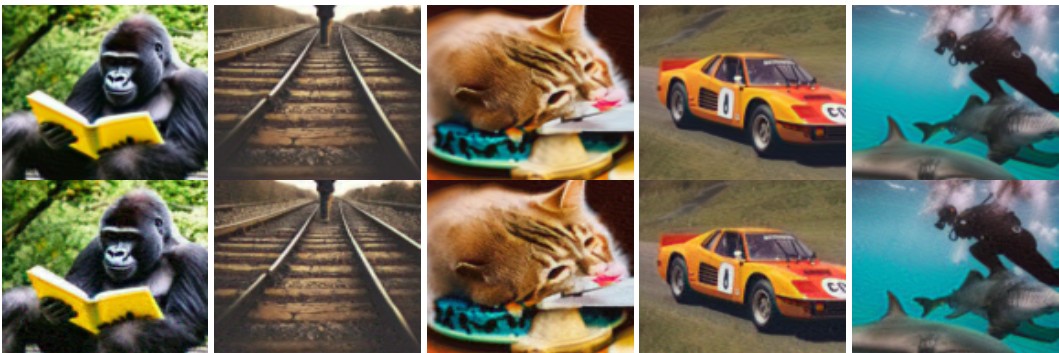

Figure 1: Watermarked images generated by Stable Diffusion (first row) and their perturbed versions in our transfer attack that successfully evade detection (second row). The target watermarking model uses ResNet architecture. Our transfer attack uses 100 surrogate watermarking models, each of which uses CNN architecture.

instead of one. Specifically, an attacker trains multiple surrogate watermarking models with different architectures and watermark lengths on a *surrogate dataset*, which may differ in distribution from the one used for the target watermarking model.

A key challenge for our attack is aggregating multiple surrogate watermarking models to find a small perturbation that evades the target watermarking model. We show that simple aggregation methods achieve suboptimal attack effectiveness. To address this, we propose a two-step approach. First, the attacker selects a *target watermark* for each surrogate model to guide the perturbation search, aiming to make each model decode this target watermark from the perturbed image. For instance, one method flips each bit of the watermark decoded by a surrogate model and uses this as the target. If multiple surrogate models decode flipped watermarks, the target model is likely to do the same, thus evading detection.

In the second step, we generate a perturbation by aggregating multiple surrogate watermarking models. A simple approach is to apply a white-box attack (Jiang et al., 2023) on each model and its target watermark, then average the resulting perturbations. However, this method performs poorly, as aggregation compromises the perturbation patterns. To overcome this, we ensemble the models and formulate an optimization problem that finds a minimum perturbation while ensuring each surrogate model decodes its target watermark. Since this problem is hard to solve, we introduce strategies to reformulate and approximate a solution.

We theoretically analyze the transferability of our attack. First, we quantify the correlation between the target and surrogate watermarking models. Using this correlation, we derive the probability that the target model's decoded watermark is flipped after adding our perturbation. From this, we further derive upper and lower bounds for the probability that the decoded watermark matches the ground truth. These bounds quantify the transferability of our attack.

We empirically evaluate our transfer attack on image datasets from Stable Diffusion and Midjourney, using multiple watermarking methods (Zhu et al., 2018; Tancik et al., 2020; Fernandez et al., 2023; Jiang et al., 2024). Our attack, using dozens of surrogate models, successfully evades watermark detectors while maintaining image quality (see examples in Figure 1). This holds even when surrogate models differ from the target in algorithms, architectures, watermark lengths, and training datasets. Our attack also outperforms common post-processing, existing transfer attacks (Jiang et al., 2023; An et al., 2024), and the state-of-the-art purification method (Nie et al., 2022), showing that existing image watermarks are broken even in the no-box setting. We note that the effectiveness of our attack to a completely new target watermarking method is unclear, which we discuss in Section 7.

To summarize, our contributions are as follows:

- We propose a transfer attack based on multiple surrogate watermarking models to watermark-based AI-generated image detector.

- We theoretically analyze the effectiveness of our attack.

- We empirically evaluate our attack and compare it with existing ones in different scenarios.

## 2 RELATED WORK

### 2.1 IMAGE WATERMARKS

**Three components:** An image watermarking method consists of three components: a *watermark* (a bitstring), an *encoder* that embeds it into an image, and a *decoder* that extracts it. In *non-learning-based methods* (Al-Haj, 2007; Bi et al., 2007; Pereira & Pun, 2000; Kang et al., 2010; Pramila et al., 2018), used for decades, the encoder and decoder are handcrafted but lack robustness to post-processing like JPEG compression or Gaussian noise (Jiang et al., 2023) (as confirmed in Section 7). In contrast, *learning-based methods* (Zhu et al., 2018; Zhang et al., 2020; Tancik et al., 2020) use deep learning, with both the encoder and decoder trained end-to-end. The encoder combines watermark and image features to generate a watermarked image, while the decoder retrieves the watermark. Joint training minimizes visual differences while ensuring accurate decoding. Learning-based methods, strengthened by adversarial training (Zhu et al., 2018), offer greater robustness, so we focus on them.

**Adversarial training:** Adversarial training (Madry et al., 2018; Goodfellow et al., 2015) is a standard method to train robust classifiers and has been extended to train robust watermarking models (Zhu et al., 2018). The key idea is to add a *post-processing layer* between the watermark encoder and decoder. The post-processing layer aims to mimic post-processing that a watermarked image may undergo. Specifically, the post-processing layer post-processes a watermarked image before sending it to the decoder. After jointly training the encoder and decoder using adversarial training, the decoder can still decode the watermark in a watermarked image even if it undergoes some post-processing. Thus, we use adversarial training in our experiments to train encoders and decoders.

### 2.2 EVASION ATTACKS

**White-box:** Jiang et al. (2023) proposed a white-box attack which assumes the attacker has access to the target watermark decoder. Given the target watermark decoder and a watermarked image, an attacker finds a small perturbation such that the watermark decoded from the perturbed image is close to a random watermark. In other words, the perturbation removes the watermark from the watermarked image and thus the perturbed image evades watermark-based detection. Specifically, the attacker finds the perturbation by solving an optimization problem via gradient descent.

**Black-box:** In black-box setting, an attacker has access to the API of a watermark-based detector, which detects images as AI-generated (watermarked) or non-AI-generated (non-watermarked). A black-box attack (Jiang et al., 2023; Lukas et al., 2024) uses the API to modify a watermarked image to remove its watermark. Starting with an initially perturbed image detected as non-AI-generated, the attacker gradually reduces the perturbation by repeatedly querying the API. With enough queries, the attacker can find a minimally perturbed image that evades detection.

**No-box:** In this setting, an attacker even has no access to the detection API. In such setting, *common post-processing* or *transfer attacks* can be used to remove watermarks. Specifically, common post-processing refers to image editing methods such as JPEG compression and Gaussian noise. Watermarks embedded by non-learning-based methods can be removed by common post-processing, but learning-based methods are more robust due to adversarial training (Zhu et al., 2018).

Existing transfer attacks rely on a single surrogate model (Jiang et al., 2023) or conventional classifier-based adversarial examples (An et al., 2024), but achieve limited success against learning-based watermarks (Jiang et al., 2023; An et al., 2024), even if we leverage state-of-the-art multiple surrogate classifiers based transfer adversarial examples (Chen et al., 2024). Note that compared to the existing watermark-based transfer attack (Jiang et al., 2023) that leverages only one surrogate watermarking model, our transfer attack leverages multiple surrogate watermarking models. One technical contribution of our work is to develop methods to aggregate multiple surrogate watermarking models when perturbing a watermarked image. Another technical contribution is that we theoretically analyze the effectiveness of watermark-based transfer attacks.

## 3 PROBLEM FORMULATION

### 3.1 WATERMARK-BASED DETECTION

A GenAI service provider trains a watermark encoder (*target encoder*) and decoder (*target decoder*, denoted $T$) for AI-generated image detection. A watermark $w$ is embedded into each image during generation and decoded from an image $x$ using $T(x)$. The *bitwise accuracy $BA(w_1, w_2)$* measures the proportion of identical bits between two watermarks $w_1$ and $w_2$. Detection determines if an image is AI-generated based on the bitwise accuracy $BA(T(x), w)$ between the decoded watermark and the ground-truth watermark. Specifically, $x$ is flagged as AI-generated if $BA(T(x), w)$ exceeds a threshold $\tau$ or falls below $1 - \tau$. $\tau$ is set to ensure the false detection rate does not exceed a small value $\eta$ (Jiang et al., 2023) (e.g., for a 30-bit watermark and $\eta = 10^{-4}$, $\tau \approx 0.83$).

The above is known as a *double-tail detector* (Jiang et al., 2023). In a *single-tail detector*, an image is detected as AI-generated if $BA(T(x), w) > \tau$. The double-tail detector is more robust; if a perturbed watermarked image evades the double-tail detector, it will evade the single-tail detector as well, but not vice versa. Thus, we focus on the double-tail detector in this work.

### 3.2 THREAT MODEL

**Attacker's goal:** Suppose an attacker uses a GenAI service to produce a watermarked image $x_w$. The attacker's goal is to introduce a minimal perturbation $\delta$ to the watermarked image $x_w$, aiming to evade watermark-based detection while preserving image quality. Consequently, the attacker can engage in illicit activities using this image, such as boosting disinformation and propaganda campaigns as well as claiming ownership of the image.

**Attacker's knowledge:** A GenAI service provider's watermark-based detector includes a target encoder, decoder, ground-truth watermark $w$, and a detection threshold $\tau$. In a *no-box* setting, the attacker has no access to any of these components, nor the architecture of the encoder/decoder, the length of $w$, or the dataset used for training. This threat model applies when the GenAI service keeps its detector private and limits API access to trusted customers.

**Attacker's capability:** We assume an attacker can add a perturbation to an AI-generated, watermarked image. Furthermore, we assume that the attacker has sufficient computational resources to train multiple surrogate watermarking models, where each surrogate watermarking model includes a surrogate encoder and a surrogate decoder.

## 4 OUR TRANSFER ATTACK

### 4.1 OVERVIEW

We propose a transfer attack to evade watermark-based AI image detection by training multiple surrogate watermarking models. These models, trained independently on a different dataset than the target model, are used to generate perturbations for watermarked images. For a given image $x_w$, we generate a perturbation $\delta$ such that the watermark decoded by each surrogate model for $x_w + \delta$ differs significantly from the original watermark. The intuition is that if multiple surrogate decoders produce different watermarks, the target decoder will likely do the same, evading detection. The same surrogate decoders are used for all images. Next, we detail the training of surrogate models and perturbation generation.

### 4.2 TRAIN SURROGATE WATERMARKING MODELS

Transfer attacks require surrogate watermarking models to generate perturbations for watermarked images. A key challenge is ensuring diversity among these models to improve transferability. The attacker collects a *surrogate dataset*, potentially with a different distribution from the target model's dataset, and uses *bootstrapping* (Efron & Tibshirani, 1994) to promote diversity. Specifically, $m$ subsets are resampled from the dataset, each used to train one of the $m$ surrogate models. Additionally, the attacker can vary neural network architectures and watermark lengths across the models.

## 4.3 FORMULATE AN OPTIMIZATION PROBLEM

To evade watermark-based detection, we generate a perturbation $\delta$ for a given watermarked image $x_w$ based on the $m$ surrogate decoders since the detection process only involves decoders. Specifically, our goal is to add a perturbation $\delta$ to the watermarked image $x_w$ such that, for each surrogate decoder, the decoded watermark from the perturbed image $x_w + \delta$ matches an attacker-chosen watermark, which we call *target watermark*. Our transfer attack faces two key challenges: 1) how to select a target watermark for a surrogate decoder, and 2) how to generate a perturbation $\delta$ based on the $m$ target watermarks and surrogate decoders. We discuss how to address the two challenges in the following.

**Select a target watermark for a surrogate decoder:** We use $w_i^t$ to denote the target watermark for the $i$th surrogate decoder, where $i = 1, 2, \cdots, m$. We consider the following three approaches to select $w_i^t$.

**Random-Different (RD).** This method involves randomly generating different target watermarks for the $m$ surrogate decoders. For the $i$th surrogate decoder, each bit of $w_i^t$ is sampled from $\{0, 1\}$ uniformly at random. The intuition of this method is that, if the $m$ surrogate decoders decode random watermarks from the perturbed image, then the target decoder is also likely to decode a random watermark from the perturbed image. As a result, the bitwise accuracy between the watermark $T(x_w + \delta)$ decoded by the target decoder for the perturbed image and the ground-truth watermark $w$ is expected to approach 0.5, thereby evading detection.

**Random-Same (RS).** This method randomly generates one random target watermark $w^t$ for all $m$ surrogate decoders, i.e., $w_i^t = w^t, \forall i = 1, 2, \cdots, m$. The intuition is that it is more likely to find a perturbation $\delta$ such that the $m$ surrogate decoders decode the same target watermark from the perturbed image, and thus it is more likely for the target decoder to decode $w^t$ from the perturbed image. Since $w^t$ is picked uniformly at random, the bitwise accuracy between $w^t$ and $w$ is expected to approach 0.5 and detection is evaded.

**Inverse-Decode (ID).** Generating random watermarks for surrogate decoders poses a challenge, as about 50% bits must be flipped during perturbation optimization, given that each bit of the target watermark is uniformly sampled from $\{0, 1\}$. Some bits may be resistant to pixel changes, requiring large perturbations.

To address this, we propose using the inverse of the watermark decoded by each surrogate decoder as the target watermark. Let $S_i$ denote the $i$th surrogate decoder, which decodes a watermark from a watermarked image $x_w$, denoted as $S_i(x_w)$. We use the inverse, $1 - S_i(x_w)$, as the target watermark $w_i^t$ for $S_i$. The intuition is that if the surrogate-decoded watermark is reversed, the target decoder's output $T(x_w + \delta)$ will likely also be close to the inverse of $T(x_w)$, making the bitwise accuracy between $T(x_w + \delta)$ and $w$ very small. We note that such bitwise accuracy may be even smaller than $1 - \tau$, which means that the double-tail detector can still detect the perturbed image as AI-generated, watermarked image. We mitigate this issue using early stopping, as discussed in Section 4.4.

With the inverse watermark as target watermark, it may result in those bits that are less robust to pixel changes being flipped first during perturbation optimization process, which can lead to a smaller perturbation compared to random target watermark when the same portion of bits are flipped, as shown in our experiments.

**Aggregate $m$ surrogate decoders:** Given a target watermark $w_i^t$ for the $i$th surrogate decoder, the attacker then finds a perturbation $\delta$ for the watermarked image $x_w$ by aggregating the $m$ surrogate decoders. We consider two ways to aggregate the $m$ surrogate decoders.

**Post-Aggregate (PA).** A simple approach is to apply an existing white-box attack to find a perturbation $\delta_i$ for each surrogate decoder. For example, given a watermarked image $x_w$, the $i$th surrogate decoder $S_i$, and target watermark $w_i^t$, the attacker can use the white-box attack (Jiang et al., 2023) to obtain $\delta_i$. This yields $m$ perturbations $\{\delta_i\}_{i=1}^m$, which are then aggregated into a final perturbation $\delta$ using a method like mean or median. However, aggregation can disrupt effective patterns in the individual perturbations, limiting transferability to the target decoder, as shown in our experiments.

**Ensemble-Optimization (EO).** To address the challenge of PA, EO considers the $m$ surrogate decoders when optimizing the perturbation $\delta$. Specifically, EO aims to find a minimum perturbation such that the watermark $S_i(x_w + \delta)$ decoded by each surrogate decoder for the perturbed image

$x_w + \delta$ is the same as its corresponding target watermark $w_i^t$. Formally, we formulate an optimization problem as follows:

$$\min_{\delta} \|\delta\|_\infty \quad s.t. \ S_i(x_w + \delta) = w_i^t, \forall i = 1, 2, \cdots, m \tag{1}$$

EO finds a single perturbation $\delta$ that makes all $m$ surrogate decoders produce their respective target watermarks for the perturbed image $x_w + \delta$. This unified perturbation improves transferability to the target decoder, as demonstrated in our experiments. We use the $\ell_\infty$-norm to measure perturbation size, but our method adapts to other metrics like $\ell_2$ or SSIM, as shown in our results.

## 4.4 Solve the Optimization Problem

Solving the optimization problem in Equation 1 gets the perturbation $\delta$ for the watermarked image $x_w$. However, since the constraints of the optimization problem are extremely strict, it is difficult to solve the optimization problem. To address this challenge, we relax the constraints and reformulate the optimization problem as follows:

$$\min_{\delta} \|\delta\|_\infty \quad s.t. \ l(S_i(x_w + \delta), w_i^t) < \epsilon', \forall i = 1, 2, \cdots, m, \tag{2}$$

where $l(\cdot, \cdot)$ denotes a metric to measure the distance between two watermarks. For instance, $l(\cdot, \cdot)$ is the mean square error in our experiments. The reformulated optimization problem is still challenging to solve due to the high non-linearity of the relaxed constraints. To address this challenge, we further reformulate the optimization problem as follows:

$$\min_{\delta} \frac{1}{m} \sum_{i=1}^{m} l(S_i(x_w + \delta), w_i^t)$$
$$s.t. \|\delta\|_\infty < r, \tag{3}$$
$$\frac{1}{m} \sum_{i=1}^{m} BA(S_i(x_w + \delta), w_i^t) > 1 - \epsilon,$$

where $r$ is a perturbation budget and $\epsilon$ (called *sensitivity*) is used to ensure that the bitwise accuracy between the watermark decoded by each surrogate decoder for the perturbed image and the corresponding target watermark is high enough to produce an effective perturbation.

We use projected gradient descent (PGD) (Madry et al., 2018) to solve the optimization problem in Equation 3. To control the strength of our transfer attack, we apply the second constraint as an early stopping condition. Algorithm 1 (Appendix) outlines the procedure for finding the perturbation $\delta$. The algorithm initializes $\delta$ to zero, computes gradients of the objective in Equation 3, and updates $\delta$ using gradient descent. If the $\ell_\infty$-norm of $\delta$ exceeds $r$, it is projected back onto the $\ell_\infty$-norm ball. The algorithm stops when a set iteration limit or bitwise accuracy threshold is reached.

## 5 Theoretical Analysis

Given the watermarks decoded by the $m$ surrogate decoders for the perturbed image, we derive both an upper bound and a lower bound for the probability that the watermark decoded by the target decoder for the perturbed image matches with the ground-truth watermark, which quantify the transferability of our transfer attack. Specifically, we derive the following theorem. Other details are shown in Appendix B.

**Theorem 1.** *For any watermarked image $x_w$, perturbation $\delta$ satisfying the constraints in Equation 3, and watermarks decoded by $m$ surrogate decoders for the perturbed image, the probability that the $j$th bit of the watermark decoded by $T$ matches the ground-truth watermark $w$ is bounded as follows:*

$$Pr(T(x_w + \delta)_j = w_j \mid S_1(x_w + \delta)_j, \cdots, S_m(x_w + \delta)_j) \geq \begin{cases} \max(\beta_j - p_j, 0), & 1 \leq j \leq n, \\ 0, & n < j \leq n_t. \end{cases}$$

$$Pr(T(x_w + \delta)_j = w_j \mid S_1(x_w + \delta)_j, \cdots, S_m(x_w + \delta)_j) \leq \begin{cases} 1 - |p_j + \beta_j - 1|, & 1 \leq j \leq n, \\ 1, & n < j \leq n_t. \end{cases}$$

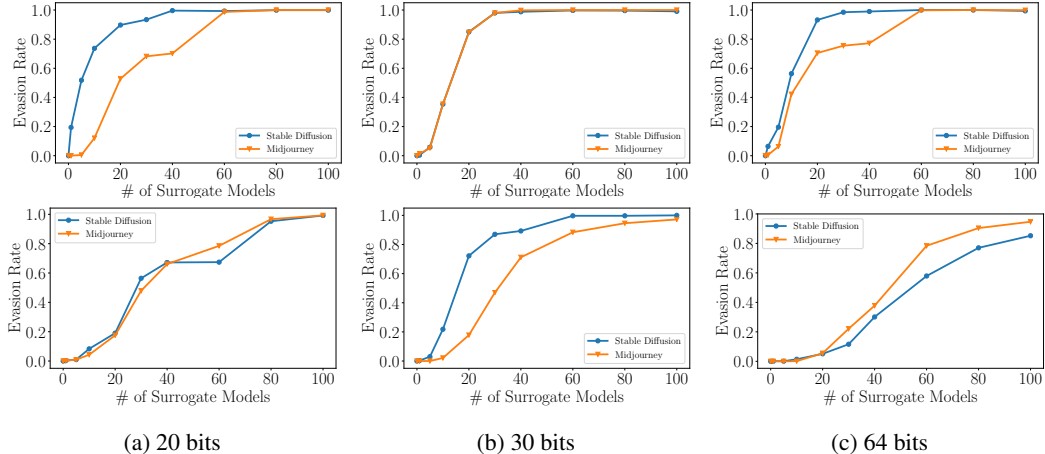

Figure 2: Evasion rate of our transfer attack when the target model uses CNN (*first row*) or ResNet (*second row*) architecture and different watermark lengths (*the three columns*). The surrogate models use CNN architecture and watermark length of 30 bits.

# 6 EXPERIMENTS

## 6.1 EXPERIMENTAL SETUP

**Datasets:** In our experiments, we utilize three publicly available datasets (Wang et al., 2023; Turc & Nemade, 2022; Images, 2023) generated by Stable Diffusion, Midjourney, and DALL-E 2. The first two datasets are used to train and test the target watermarking models, while the last dataset is used to train the surrogate watermarking model. Each training set contains 10,000 images, and each testing set contains 1,000 images. The details of the datasets are introduced in Appendix H.

**Surrogate watermarking models and watermark lengths:** Unless otherwise mentioned, we use HiDDeN (Zhu et al., 2018) and its CNN architecture with a 30-bit watermark length to train surrogate watermarking models. This consistent setup serves two purposes: 1) it helps clearly assess our attack's transferability when the target and surrogate models have different architectures or watermark lengths, and 2) it allows for better analysis of how the number of surrogate models affects transferability, without the confounding factors of varying architectures or watermark lengths.

**Target watermarking models and watermark lengths:** For the target watermarking model, we use different choices of watermarking methods, architectures, and watermark lengths to analyze the transferability of our transfer attack in different scenarios. Specifically, we consider HiDDeN (Zhu et al., 2018), StegaStamp (Tancik et al., 2020), Stable Signature (Fernandez et al., 2023), Smoothed HiDDeN (Jiang et al., 2024), and Smoothed StegaStamp (Jiang et al., 2024) to train target watermarking models. The detailed settings of each watermarking method are shown in Appendix I.

**Common post-processing methods:** We consider 4 common post-processing methods. They are JPEG, Gaussian noise, Gaussian blur, and Brightness/Contrast. Details are introduced in Appendix J.

**Transfer attacks:** In our experiments, we compare with 5 existing transfer attacks and 1 state-of-the-art purification method for adversarial examples. They are WEvade-B-S (Jiang et al., 2023), AdvEmb-RN18 (An et al., 2024), AdvCls-Real&WM (An et al., 2024), AdvCls-Enc-WM1&WM2, MI-CWA (Chen et al., 2024), and DiffPure (Nie et al., 2022). Details are introduced in Appendix K.

**Evaluation metrics:** We use two metrics: *evasion rate* and *average perturbation*. Evasion rate is the proportion of perturbed images that evade the target watermark detector. Average perturbation, measured by the $\ell_\infty$-norm, is averaged across 1,000 watermarked images in the test set. Following Jiang et al. (2023), with pixel values normalized from [0, 255] to [-1, 1], we divide the perturbation by 2 to express it as a fraction of the full pixel range.

**Parameter settings:** We use adversarial training to train watermarking models since it enhances robustness. The detailed settings of parameters are shown in Appendix L

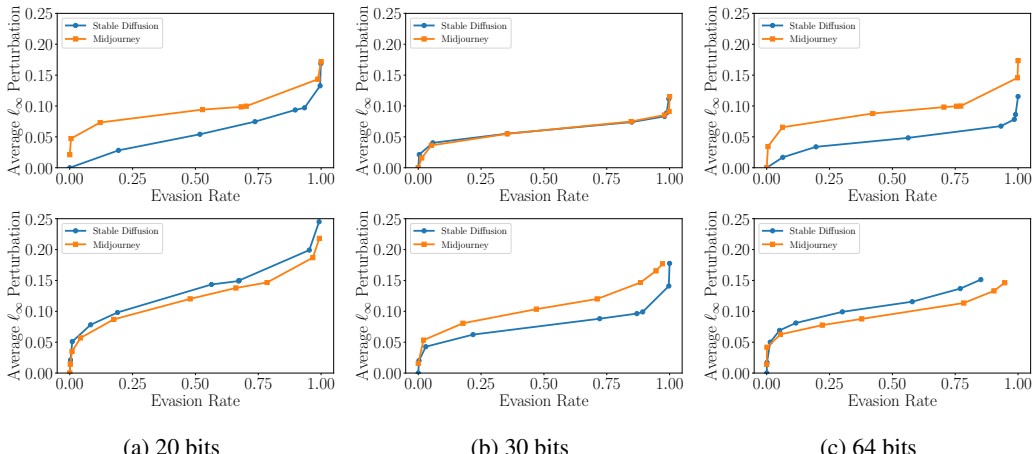

|(a) 20 bits|(b) 30 bits|(c) 64 bits|

Figure 3: Average perturbation of our transfer attack when the target model uses CNN (*first row*) or ResNet (*second row*) architecture and different watermark lengths (*the three columns*). The surrogate models use CNN architecture and watermark length of 30 bits.

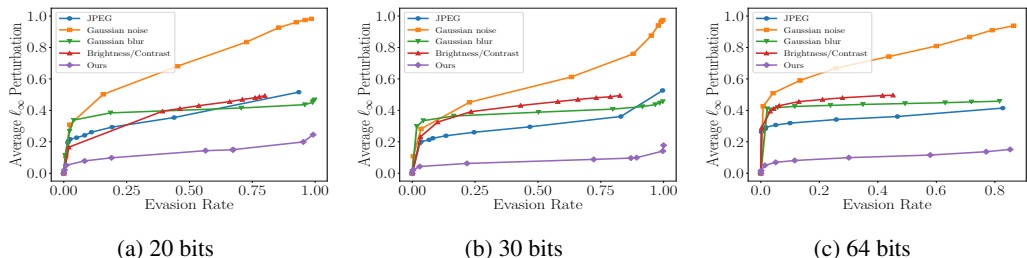

|(a) 20 bits|(b) 30 bits|(c) 64 bits|

Figure 4: Comparing the average perturbations of our transfer attack and common post-processing methods when they achieve the same evasion rate. The target model uses ResNet architecture and different watermark lengths (*the three columns*). Dataset is Stable Diffusion. Results for Midjourney are shown in Figure 7 in Appendix.

## 6.2 EXPERIMENTAL RESULTS

**Our transfer attack is successful:** Figure 2 and Figure 3 show the evasion rate and average perturbation of our attack, across different numbers of surrogate models, target model architectures, and watermark lengths. First, our attack successfully evades watermark-based detection, achieving an evasion rate over 85% with an average perturbation below 0.25 when using 100 surrogate models, regardless of target architecture or watermark length. Second, more complex target architectures require more surrogate models and higher perturbation. For example, 40 surrogate models and perturbation of 0.09 achieve 100% evasion for a CNN with 30-bit watermarks, while 60 models and 0.14 perturbation are needed for a ResNet. Third, longer/shorter watermarks also increase the attack's complexity. For a ResNet with 30-bit watermarks, 40 surrogate models and perturbation of 0.08 yield 85% evasion, while 100 models and 0.15 perturbation are needed for 64-bit watermarks.

**Our transfer attack outperforms common post-processing:** Figure 4 compares the average perturbations of common post-processing methods and our transfer attack when they achieve the same evasion rates for different target models. For each evasion rate achieved by our transfer attack, we adjust the parameters of the post-processing methods to match it. We find that our attack introduces much smaller perturbations than post-processing methods for the same evasion rate. Note that Brightness/Contrast only reaches up to 75% or 50% evasion for some models, so its curves are shorter in the graphs. For a broader comparison, Figures 8 and 9 in Appendix show results using $\ell_2$-norm and SSIM. Our attack consistently introduces smaller perturbations and maintains better visual quality than common post-processing, regardless of the metric used.

**Our transfer attack outperforms existing ones:** Figure 5a compares the evasion rates of existing methods and our transfer attack when the target model is ResNet with varying watermark lengths. Since AdvCls-Real&Wm and AdvCls-Enc-WM1&WM2 achieve average perturbations around 0.1 regardless of the budget $r$, we introduce a variant of our method with $r = 0.1$ for comparison. For fairness, we constrain MI-CWA and DiffPure to an average SSIM of 0.9, similar to our attack with

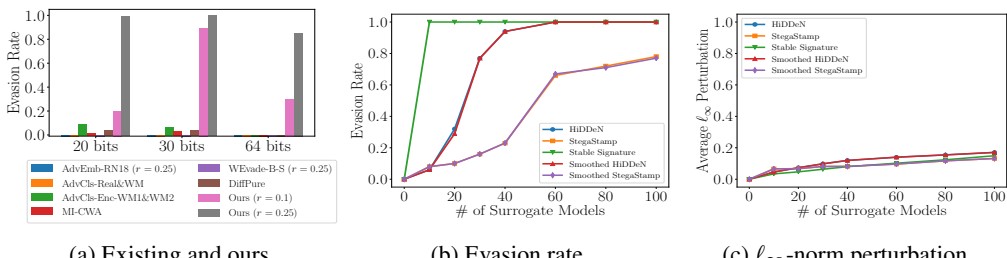

(a) Existing and ours      (b) Evasion rate      (c) $\ell_\infty$-norm perturbation

Figure 5: (a) Comparing evasion rates of existing and our transfer attacks. The target model is ResNet and uses watermarks with different lengths. Dataset is Stable Diffusion. Similar results for Midjourney are shown in Figure 10 in Appendix. (b) Evasion rates and (c) average $\ell_\infty$-norm perturbation of our transfer attacks to different target watermarking methods.

100 surrogate models. Other attacks have $r$ set to 0.25. We also report the evasion rate for each method across a wide range of SSIM constraints in Figure 11 in Appendix.

Our transfer attack significantly outperforms existing methods, which mostly achieve 0% evasion, except AdvCls-Enc-WM1&WM2, which reaches 10% on Stable Diffusion with 20-30 bit watermarks but requires access to the target encoder. MI-CWA and DiffPure show low evasion rates due to the limitations of classifier-based attacks and heavy image modification for watermark removal. In contrast, our transfer attack achieves much higher evasion rates while maintaining high image quality, with an average SSIM above 0.97 for $r = 0.1$ and above 0.92 for $r = 0.25$.

**Our transfer attack transfers to other target watermarking methods:** Figures 5b and 5c show the evasion rates and average $\ell_\infty$-norm perturbations of our transfer attack across different target watermarking methods. To improve transferability, we use equal numbers of HiDDeN and StegaStamp as surrogate models. Our attack remains effective even when surrogate and target methods differ, achieving over 77% evasion across all five target methods with 100 surrogate models. Despite Smoothed HiDDeN and Smoothed StegaStamp being certifiably robust, they are still vulnerable to our attack. While evasion rates are slightly lower for Smoothed HiDDeN with 20 models and Smoothed StegaStamp with over 60 models, we still exceed 77% evasion with up to 100 models, as our perturbations exceed their certified bounds while preserving image quality.

**Impact of sensitivity $\epsilon$:** Figures 12a and 12b in Appendix show the evasion rate and average $\ell_\infty$-norm perturbation for different sensitivity values ($\epsilon$) in our transfer attack on Stable Diffusion, with a CNN target model using 30-bit watermarks. Figures 13a and 13b in Appendix show results for $\ell_2$-norm and SSIM. We observe that fewer surrogate models are needed for the same evasion rate when $\epsilon$ is smaller. However, with very small $\epsilon$ (e.g., 0.1), the evasion rate first increases but then decreases as more surrogate models are used, due to reduced bitwise accuracy in watermark decoding. Across different $\epsilon$ values, the average perturbation (whether measured by $\ell_\infty$-norm, $\ell_2$-norm, or SSIM) remains nearly constant for a given evasion rate, reflecting the strong correlation between perturbation size and evasion success.

**Different variants of our transfer attack:** Our attack has two steps, each with multiple design options. In the first step, we use RD, RS, or ID; in the second step, PA or EO is used. Variants are denoted by combining these symbols, e.g., RD-PA. When PA is used, "Mean" or "Median" indicates the aggregation rule, such as RD-PA-Mean. Figures 12c and 12d in Appendix show evasion rates and $\ell_\infty$-norm perturbations for different variants on Stable Diffusion, with a CNN using 30-bit watermarks. Figures 13c and 13d in Appendix show $\ell_2$-norm and SSIM results. EO-based variants successfully evade detection, while PA-based variants do not, showing EO is more effective for surrogate decoder aggregation. Among EO variants, ID-EO outperforms RD-EO and RS-EO, achieving 100% evasion with fewer surrogate models, lower perturbations, and higher SSIM.

**Theoretical vs. empirical results:** Figures 6a and 6b show the empirical bitwise accuracy between the watermark decoded by the target decoder and the ground truth, along with our theoretical upper and lower bounds on the Stable Diffusion dataset. To estimate these bounds, we measured unperturbed similarity, transfer similarity, and attack strength using 1,000 test images. Using Theorem 1, we computed the probability for each bit $p_j$, then averaged the bounds across all bits. We observe that the empirical results generally fall within our theoretical bounds, with a few exceptions due to the use of only 1,000 images for estimation, leading to minor inaccuracies.

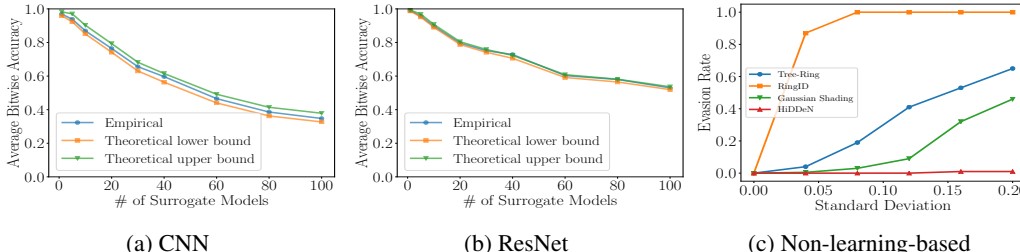

Figure 6: Comparing empirical bitwise accuracy and theoretical bounds of our transfer attack when the target model is (a) CNN and (b) ResNet. (c) Comparing evasion rates of non-learning-based watermarking models and HiDDeN under Gaussian noise with varying standard deviations.

## 7 DISCUSSION AND LIMITATIONS

**Non-learning-based watermarking:** We focus on transfer attacks against learning-based watermarking models because they are more robust than non-learning-based ones. For example, common post-processing like Gaussian noise can easily remove watermarks from non-learning-based models, but learning-based models resist such attacks (Jiang et al., 2023). Figure 6c presents the evasion rates of Tree-Ring (Wen et al., 2024), RingID (Ci et al., 2025), Gaussian Shading (Yang et al., 2024), and HiDDeN with ResNet under Gaussian noise. For the standard deviation larger than 0.1, the non-learning-based methods exhibit significantly higher evasion rates, whereas the learning-based method maintains an evasion rate close to zero. These results underscore the superior robustness of learning-based watermarking methods compared to the non-learning-based ones.

Our attack can also be applied to non-learning-based methods. Figure 15 in the Appendix shows that our attack achieves high evasion rates when the target watermarking method is DWT-DCT (Al-Haj, 2007) while the surrogate models are trained under our default experimental settings. Unlike learning-based methods, DWT-DCT is a non-learning-based watermarking method used by Stable Diffusion. It employs the Discrete Wavelet Transform (DWT) to decompose an image into frequency sub-bands, applies the Discrete Cosine Transform (DCT) to blocks within selected sub-bands, and embeds the watermark by modifying specific frequency coefficients. The watermarked image is then reconstructed using inverse transforms. The result shows that our attack requires more surrogate models compared to learning-based methods. This is because non-learning-based watermarking method is substantially different from learning-based, and our attack needs more surrogate models to increase diversity to enhance transferability.

We acknowledge that the victim may design a new watermarking method entirely different from those used by the attacker as surrogates, and the effectiveness of our attack in such scenarios is unclear. We believe this can be an interesting future work to explore.

**Computational cost:** Our transfer attack trains surrogate watermarking models and optimizes perturbations for watermarked images. Thus, the computational cost consists of training time (to train surrogate models) and inference time (to optimize a perturbation for an image). Table 2 in Appendix compares the computational costs of our attack with six baselines. Attacking high-capacity watermarks (Zhong et al., 2020) may require more surrogate models, further increasing training cost—a practical limitation of our approach. However, training multiple surrogate models can be parallelized across GPUs, and this training is a one-time process, which helps mitigate the limitation. We acknowledge that our inference time exceeds most baselines but remains acceptable with adequate computational resources.

## 8 CONCLUSION AND FUTURE WORK

In this work, we find that watermark-based detection of AI-generated images is not robust to transfer attacks in the no-box setting. Given a watermarked image, an attacker can remove the watermark by adding a perturbation to it, where the perturbation can be found by ensembling multiple surrogate watermarking models. Our results show that transfer attack based on surrogate watermarking models outperforms those based on surrogate classifiers that treat a watermark-based detector as a conventional classifier. Moreover, leveraging surrogate watermarking models enables us to perform a rigorous analysis on the transferability of our attack. Interesting future work includes extending our transfer attack to text and audio watermarks, as well as designing more robust watermarks.

## 9 ACKNOWLEDGMENT

We sincerely thank all the reviewers for their constructive feedback. This work was supported by NSF grants No. 2414406, 2131859, 2125977, 2112562, and 1937787, as well as ARO grant No. W911NF2110182.

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

Table 1: The detailed settings of $\alpha$ for different number of surrogate models.

| # of Surrogate Models | 1 | 5 | 10 | 20 | 30 | 40 | 60 | 80 | 100 |
|---|---|---|---|---|---|---|---|---|---|
| $\alpha$ | | 0.1 | | | 1 | | 2 | | 4 |

Table 2: Computational cost comparison of existing attacks and our transfer attack on a single NVIDIA RTX-6000 GPU. $\Delta_{\mathrm{RN18}}$ denotes the pre-training time of ResNet-18 on ImageNet, and $\Delta_{\mathrm{Diff}}$ denotes the training time of the unconditional diffusion model used by DiffPure. We note that training only needs to be done once.

| Method | Training Time (h) | Inference Time per Image (s) |
|---|---|---|
| AdvEmb-RN18 | $\Delta_{\mathrm{RN18}}$ | 72.09 |
| AdvCls-Real&WM | $\Delta_{\mathrm{RN18}} + 0.11$ | 77.04 |
| AdvCls-Enc-WM1&WM2 | $\Delta_{\mathrm{RN18}} + 0.11$ | 77.70 |
| MI-CWA | $\Delta_{\mathrm{RN18}} + 0.11$ | 27.01 |
| DiffPure | $\Delta_{\mathrm{Diff}}$ | 8.96 |
| WEvade-B-S | 17.06 | 187.03 |
| Ours | 6.50 | 177.97 |

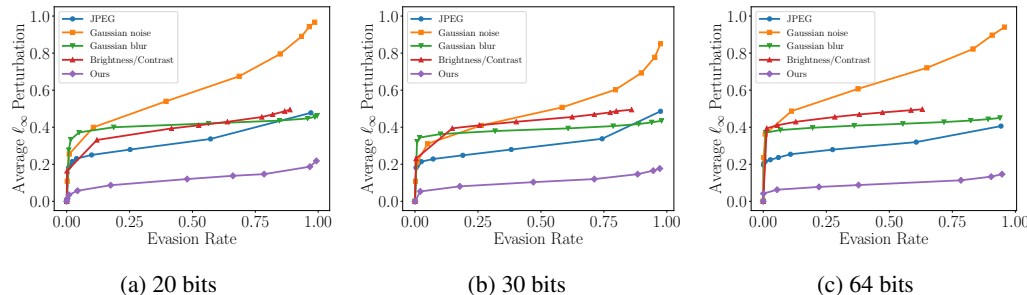

(a) 20 bits          (b) 30 bits          (c) 64 bits

Figure 7: Comparing the average perturbations of our transfer attack and common post-processing methods when they achieve the same evasion rate. The target model uses ResNet architecture and different watermark lengths (*the three columns*). Dataset is Midjourney.

## A    ETHICS CONCERNS

Our proposed method has the potential to compromise the identity and authenticity of AI-generated images, raising ethical and societal concerns. Such misuse could erode trust in digital media by enabling unauthorized modifications or misattributions. To mitigate these risks, it is crucial to develop robust watermarking methods that can withstand our transfer attack. As discussed in Section 7, our attack's effectiveness is unclear when the target watermarking method is entirely unknown and significantly different from the surrogate methods used by attackers. Therefore, designing novel watermarking techniques with fundamentally different mechanisms and maintaining their confidentiality could serve as an effective strategy to mitigate these risks.

## B    THEORETICAL ANALYSIS

We derive both an upper bound and a lower bound of the probability because we consider double-tail detectors. Specifically, they enable us to compute the upper bound and lower bound of the expected bitwise accuracy between the watermark decoded by the target decoder for the perturbed image and the ground-truth watermark. If such upper bound and lower bound of the expected bitwise accuracy fall between $\tau$ and $1 - \tau$, the perturbed image produced by our transfer attack is expected to evade the target watermark-based detector. All our proofs are shown in Appendix.

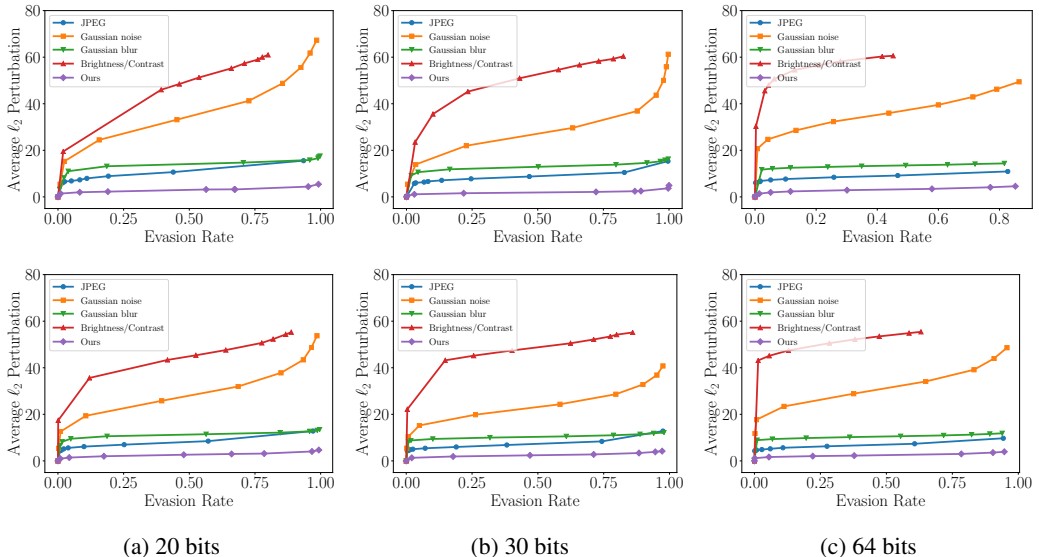

Figure 8: Comparing the average $\ell_2$-norm perturbations of our transfer attack and common post-processing methods when they achieve the same evasion rate. The target model uses ResNet architecture and different watermark lengths (*the three columns*). *First row*: Stable Diffusion. *Second row*: Midjourney.

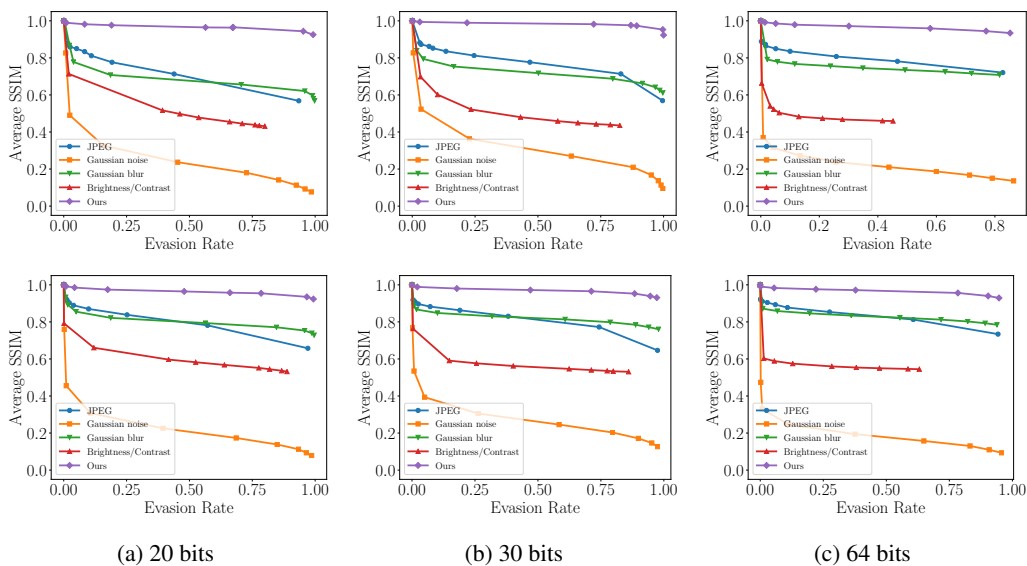

Figure 9: Comparing the average SSIM of our transfer attack and common post-processing methods when they achieve the same evasion rate. The target model uses ResNet architecture and different watermark lengths (*the three columns*). *First row*: Stable Diffusion. *Second row*: Midjourney.

## B.1 NOTATIONS

We adopt $T$ to denote the target decoder and $\{S_i\}_{i=1}^{m}$ to represent the $m$ surrogate decoders. $w$ denotes the ground-truth watermark and $x_w$ denotes a watermarked image embedded with $w$. $\delta$ denotes a perturbation that satisfies the constraints of the optimization problem in Equation 3. We use $n_s$ and $n_t$ to denote the watermark length of surrogate decoders and target decoder respectively, while we define $n = min(n_s, n_t)$. $T(\cdot)_j$ and $S_i(\cdot)_j$ denote the $j$th bit of the watermarks decoded by $T$ and $S_i$ for an image, respectively. Moreover, we consider inverse watermarks as the target watermarks in our transfer attack.

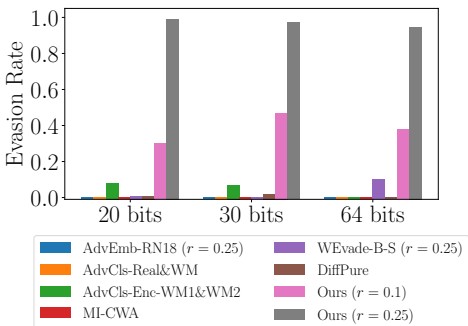

Figure 10: Comparing the evasion rates of existing and our transfer attacks when the target model is ResNet and uses watermarks with different lengths. Dataset is Midjourney.

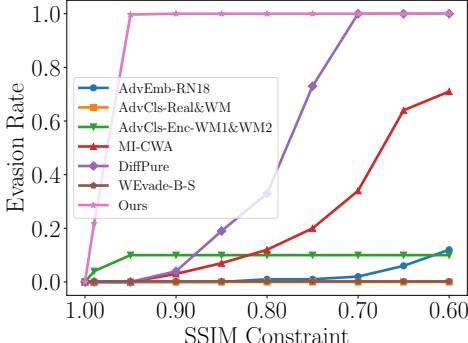

Figure 11: Comparing the evasion rates of existing and our transfer attacks under different SSIM constraints between the watermarked and perturbed images. The target model uses ResNet architecture and watermark length of 30 bits. Dataset is Stable Diffusion.

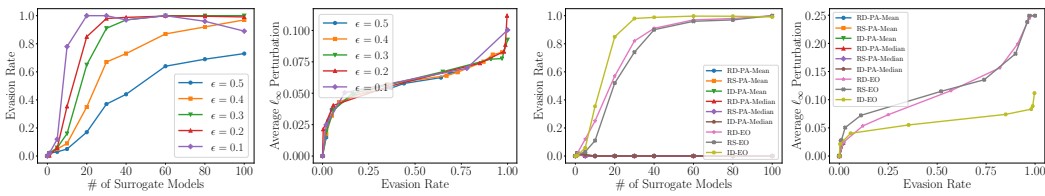

(a) Evasion rate  (b) $\ell_\infty$-norm perturbation  (c) Evasion rate  (d) $\ell_\infty$-norm perturbation

Figure 12: (a) Evasion rate and (b) average $\ell_\infty$-norm perturbation of our transfer attack when using different sensitivity $\epsilon$. (c) Evasion rate and (d) average $\ell_\infty$-norm perturbation of different variants of our transfer attack.

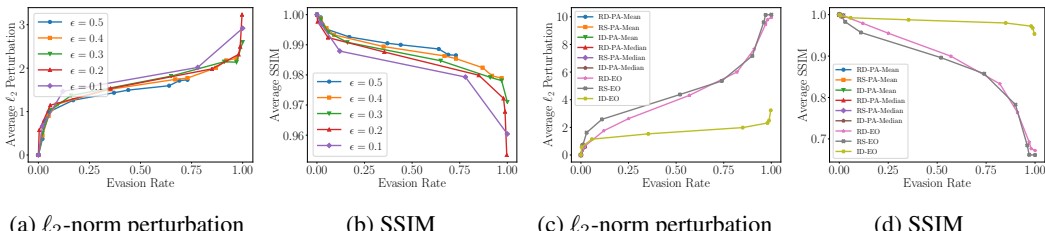

(a) $\ell_2$-norm perturbation  (b) SSIM  (c) $\ell_2$-norm perturbation  (d) SSIM

Figure 13: (a, b) Average $\ell_2$-norm perturbations and average SSIM of our transfer attack with different sensitivity $\epsilon$. The target model uses CNN architecture and watermark length of 30 bits. (c, d) Average $\ell_2$-norm perturbations and average SSIM of different variants of our transfer attack. The target model uses CNN architecture and watermark length of 30 bits.

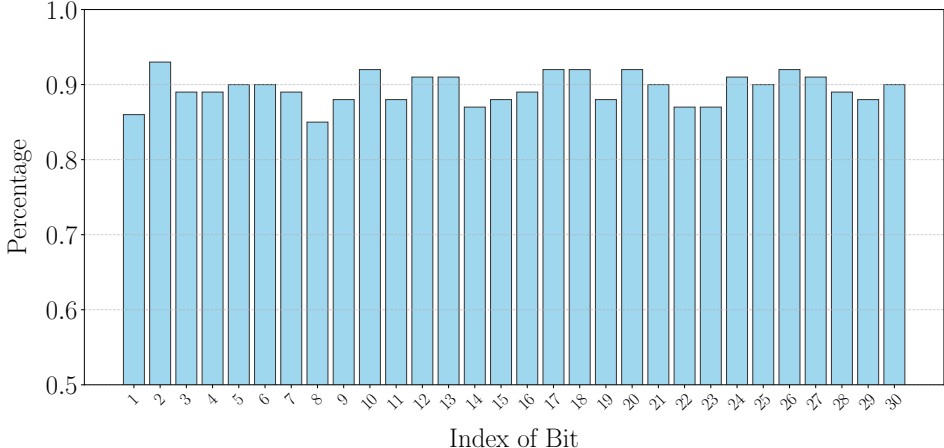

Figure 14: Percentage of the pairs of surrogate watermarking models that pass the Chi-square test of independence with confidence of 0.95 on each bit. We randomly sampled 100 pairs of surrogate watermarking models from the 100 surrogate models in our experiments. For each bit of the watermark and pair of surrogate models, we performed the Chi-square independence test with a confidence level of 0.95 using the decoded watermarks from the 1,000 perturbed images. The y-axis in the graph shows the percentage of the 100 pairs of the surrogate models that passed the independence test for each bit. These results verify that our independence assumption in Assumption 2 roughly hold.

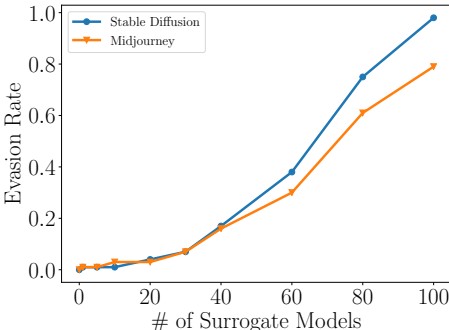

Figure 15: Evasion rate of our transfer attack when the target watermarking model is non-learning-based.

## B.2 FORMAL DEFINITIONS

In this section, we will introduce the details of our formal definitions. We formally define *unperturbed similarity*, *positive transferring similarity*, *negative transferring similarity*, *q-attacking strength*, *β-accurate watermarking*, *Bit-level dependency*, *Independency*, and *Conditional independency*. Specifically, unperturbed similarity measures the probability that the watermarks decoded by the target decoder and surrogate decoders are identical for a watermarked image $x_w$ without perturbation, which is formally defined as follows:

**Definition 1** (Unperturbed similarity). *For any watermarked image $x_w$ and $\delta$ satisfying the constraints of the optimization problem in Equation 3, the jth bit of the watermark decoded by $T$ matches with the jth bit of the watermark decoded by $S_i$ for $x_w$ with probability $k_{ij}$, given that the jth bits of the watermarks for $x_w + \delta$ and $x_w$ decoded by $S_i$ are inverse. Conversely, it occurs with probability $k'_{ij}$ given that the jth bits of the watermarks for $x_w + \delta$ and $x_w$ decoded by $S_i$ are identical. Formally, for $i = 1, 2, \cdots, m$ and $j = 1, 2, \cdots, n$, we have:*

$$
\begin{aligned}
Pr(T(x_w)_j = S_i(x_w)_j \mid S_i(x_w + \delta)_j = 1 - S_i(x_w)_j) = k_{ij}, \\
Pr(T(x_w)_j = S_i(x_w)_j \mid S_i(x_w + \delta)_j = S_i(x_w)_j) = k'_{ij}.
\end{aligned}
\tag{4}
$$

---

**Algorithm 1** Find the Perturbation $\delta$

---

**Require:** Watermarked image $x_w$, $m$ surrogate decoders $\{S_i\}_{i=1}^m$, $m$ target watermarks $\{w_i^t\}_{i=1}^m$, distance metric $l$, perturbation budget $r$, sensitivity $\epsilon$, learning rate $\alpha$, and maximum number of iterations $max\_iter$.
**Ensure:** Perturbation $\delta$
1: Initialize $\delta \leftarrow 0$
2: **for** $k = 1, 2, \cdots, max\_iter$ **do**
3:      $g \leftarrow \nabla_\delta \frac{1}{m} \sum_{i=1}^m l(S_i(x_w + \delta), w_i^t)$
4:      $\delta \leftarrow \delta - \alpha \cdot g$
5:      **if** $\|\delta\|_\infty > r$ **then**
6:          $\delta \leftarrow \delta \cdot \frac{r}{\|\delta\|_\infty}$
7:      **end if**
8:      **if** $\frac{1}{m} \sum_{i=1}^m BA(S_i(x_w + \delta), w_i^t) \geq 1 - \epsilon$ **then**
9:          Return $\delta$
10:     **end if**
11: **end for**
12: Return $\delta$

---

Note that both $k_{ij}$ and $k'_{ij}$ can be estimated in experiments. Positive transferring similarity measures the probability that the watermarks decoded by target decoder and surrogate decoders for a perturbed image $x_w + \delta$ are identical when the watermarks decoded by the surrogate decoders are flipped after adding the perturbation, which is defined as follows:

**Definition 2** (Positive transferring similarity). *For any watermarked image $x_w$ and $\delta$ satisfying the constraints of the optimization problem in Equation 3, the jth bit of the watermark decoded by $T$ matches with the jth bit of the watermark decoded by $S_i$ for $x_w + \delta$ with probability $a_{ij}$, given that the jth bits of the watermarks decoded by $T$ and $S_i$ are identical for $x_w$, and the jth bits of the watermarks for $x_w + \delta$ and $x_w$ decoded by $S_i$ are inverse. Conversely, it occurs with probability $a'_{ij}$ given that the jth bits of the watermarks decoded by $T$ and $S_i$ for $x_w$ are inverse, and the jth bits of the watermarks for $x_w + \delta$ and $x_w$ decoded by $S_i$ are also inverse. Formally, for $i = 1, 2, \cdots, m$ and $j = 1, 2, \cdots, n$, we have the following:*

$$
\begin{aligned}
Pr(T(x_w + \delta)_j =& S_i(x_w + \delta)_j \mid T(x_w)_j = S_i(x_w)_j, \\
& S_i(x_w + \delta)_j = 1 - S_i(x_w)_j) = a_{ij}, \\
Pr(T(x_w + \delta)_j =& S_i(x_w + \delta)_j \mid T(x_w)_j = 1 - S_i(x_w)_j, \\
& S_i(x_w + \delta)_j = 1 - S_i(x_w)_j) = a'_{ij}.
\end{aligned}
\tag{5}
$$

Both $a_{ij}$ and $a'_{ij}$ can be estimated in experiments. Similarly, we use negative transferring similarity to measure the probability that the watermarks decoded by target decoder and surrogate decoders for a perturbed image $x_w + \delta$ are identical when the outputs of surrogate decoders remain unchanged after adding the perturbation, which is defined as follows:

**Definition 3** (Negative transferring similarity). *For any watermarked image $x_w$ and $\delta$ satisfying the constraints of the optimization problem in Equation 3, the jth bit of the watermark decoded by $T$ matches with the jth bit of the watermark decoded by $S_i$ for $x_w + \delta$ with probability $b_{ij}$, given that the jth bits of the watermarks decoded by $T$ and $S_i$ are inverse for $x_w$, and the jth bits of the watermarks for $x_w + \delta$ and $x_w$ decoded by $S_i$ are identical. Conversely, it occurs with probability $b'_{ij}$ given that the jth bits of the watermarks decoded by $T$ and $S_i$ for $x_w$ are identical, and the jth bits of the watermarks for $x_w + \delta$ and $x_w$ decoded by $S_i$ are also identical. Formally, for $i = 1, 2, \cdots, m$ and $j = 1, 2, \cdots, n$, we have the following:*

$$
\begin{aligned}
Pr(T(x_w + \delta)_j = & S_i(x_w + \delta)_j \mid T(x_w)_j = 1 - S_i(x_w)_j, \\
& S_i(x_w + \delta)_j = S_i(x_w)_j) = b_{ij}, \\
Pr(T(x_w + \delta)_j = & S_i(x_w + \delta)_j \mid T(x_w)_j = S_i(x_w)_j, \\
& S_i(x_w + \delta)_j = S_i(x_w)_j) = b'_{ij}.
\end{aligned}
\tag{6}
$$

Both $b_{ij}$ and $b'_{ij}$ can be estimated in experiments. To quantify the magnitude of our transfer attack, we formally define the $q$-attacking strength which measures the probability that the watermarks for the perturbed image and the watermarked image decoded by surrogate decoders are inverse as follows:

**Definition 4** ($q$-attacking strength). *For any watermarked image $x_w$ and $\delta$ satisfying the constraints of the optimization problem in Equation 3, the $j$th bit of the watermark decoded by $S_i$ for the perturbed image is the inverse of the $j$th bit of the watermark decoded by $S_i$ for the watermarked image with probability $q_{ij}$. Formally, for $i = 1, 2, \cdots, m$ and $j = 1, 2, \cdots, n_s$, we have the following:*

$$Pr(S_i(x_w + \delta)_j = 1 - S_i(x_w)_j) = q_{ij}. \tag{7}$$

$q_{ij}$ can also be estimated in experiments. To quantify the performance of the target watermarking model when no perturbations are added to watermarked images, we introduce the following definition on $\beta$-accurate watermarking: assume the bitwise accuracy between the watermark decoded by the target decoder for the watermarked image and the ground-truth watermark following Poisson binomial distribution. Specifically, we have the following assumption:

**Definition 5** ($\beta$-accurate watermarking). *For any watermarked image $x_w$, the bits of the watermark decoded by $T$ are mutually independent and the probability that the $j$th bit of the decoded watermark matches with that of the ground-truth watermark $w$ is $\beta_j$, where $1 \leq j \leq n_t$.*

$\beta_1, \beta_2, \cdots, \beta_{n_t}$ can be estimated in experiments. Next, we introduce three assumptions regarding the surrogate decoders and the target decoder. For bit-level dependency, it is assumed that each bit of the watermark decoded by the target decoder only depends on the corresponding bits of the watermarks decoded by the surrogate decoders. Formally, we have the assumption as follows:

**Assumption 1** (Bit-level dependency). *For any watermarked image $x_w$ and $\delta$ satisfying the constraints of the optimization problem in Equation 3, each bit of the watermark decoded by $T$ for the perturbed image only depends on the corresponding bits of the watermarks decoded by the surrogate decoders for the perturbed image. Formally, for $j = 1, 2, \cdots, n$, we have the following:*

$$\begin{aligned} &Pr(T(x_w + \delta)_j \mid S_1(x_w + \delta), \cdots, S_m(x_w + \delta)) \\ &= Pr(T(x_w + \delta)_j \mid S_1(x_w + \delta)_j, \cdots, S_m(x_w + \delta)_j). \end{aligned} \tag{8}$$

Additionally, since all the surrogate decoders are trained independently with different subsets of data, we further assume that the watermarks decoded by the surrogate decoders are independent and conditionally independent given the watermark decoded by the target decoder for the watermarked or perturbed image. We verify the independency assumption and the results are shown in Figure 14.

**Assumption 2** (Independency). *For any watermarked image $x_w$ and $\delta$ satisfying the constraints of the optimization problem in Equation 3, the $j$th bits of the watermarks decoded by the $m$ surrogate decoders for the perturbed image are independent. Formally, for $j = 1, 2, \cdots, n_s$, we have the following:*

$$\begin{aligned} &Pr(S_1(x_w + \delta)_j, \cdots, S_m(x_w + \delta)_j) \\ &= Pr(S_1(x_w + \delta)_j) \cdots Pr(S_m(x_w + \delta)_j). \end{aligned} \tag{9}$$

**Assumption 3** (Conditional independency). *For any watermarked image $x_w$ and $\delta$ satisfying the constraints of the optimization problem in Equation 3, the $j$th bits of the watermarks decoded by the $m$ surrogate decoders for $x_w + \delta$ are independent when the $j$th bit of the watermark decoded by the target decoder for $x_w$ or $x_w + \delta$ is given. Formally, for $j = 1, 2, \cdots, n$, we have:*

$$\begin{aligned} &Pr(S_1(x_w + \delta)_j, \cdots, S_m(x_w + \delta)_j \mid T(x_w)_j) \\ &= Pr(S_1(x_w + \delta)_j \mid T(x_w)_j) \cdots \\ &\quad Pr(S_m(x_w + \delta)_j \mid T(x_w)_j), \forall j, \\ &Pr(S_1(x_w + \delta)_j, \cdots, S_m(x_w + \delta)_j \mid T(x_w + \delta)_j) \\ &= Pr(S_1(x_w + \delta)_j \mid T(x_w + \delta)_j) \cdots \\ &\quad Pr(S_m(x_w + \delta)_j \mid T(x_w + \delta)_j), \forall j. \end{aligned} \tag{10}$$

### B.3 DERIVING $Pr(T(x_w + \delta)_j = 1 - T(x_w)_j)$

We first derive the probability that the watermark decoded by the target decoder is flipped after adding the perturbation found by our transfer attack to the watermark image, conditioned on that the watermark decoded by a surrogate decoder is flipped or not after adding the perturbation. Formally, we have the following lemma:

**Lemma 1.** *For any watermarked image $x_w$ and $\delta$ satisfying the constraints of the optimization problem in Equation 3, the jth bit of the watermark decoded by $T$ for $x_w + \delta$ is the inverse of the jth bit of the watermark decoded by $T$ for $x_w$ with probability $a_{ij}k_{ij} + (1 - a'_{ij})(1 - k'_{ij})$, given that the jth bits of the watermarks for the $x_w + \delta$ and $x_w$ decoded by $S_i$ are inverse. Conversely, it occurs with probability $(1 - b'_{ij})k'_{ij} + b_{ij}(1 - k'_{ij})$ given that the jth bits of the watermarks for $x_w + \delta$ and $x_w$ decoded by $S_i$ are identical. Formally, for $j = 1, 2, \cdots, n$, we have:*

$$
\begin{aligned}
&Pr(T(x_w + \delta)_j = 1 - T(x_w)_j \mid S_i(x_w + \delta)_j = 1 - S_i(x_w)_j) \\
&= c_{ij}, \\
&Pr(T(x_w + \delta)_j = 1 - T(x_w)_j \mid S_i(x_w + \delta)_j = S_i(x_w)_j) \\
&= c'_{ij},
\end{aligned}
\tag{11}
$$

*where $c_{ij} = a_{ij}k_{ij} + (1 - a'_{ij})(1 - k_{ij})$ and $c'_{ij} = (1 - b'_{ij})k'_{ij} + b_{ij}(1 - k'_{ij})$.*

Then, we derive the unconditional probability that the watermark decoded by the target decoder is flipped after adding the perturbation to the watermarked image as follows:

**Theorem 2.** *For any watermarked image $x_w$ and $\delta$ satisfying the constraints of the optimization problem in Equation 3, the jth bit of the watermark decoded by $T$ for the perturbed image is the inverse of the jth bit of the watermark decoded by $T$ for the watermarked image with probability $c_{ij}q_{ij} + c'_{ij}(1 - q_{ij})$. Formally, for $j = 1, 2, \cdots, n$, we have the following:*

$$
Pr(T(x_w + \delta)_j = 1 - T(x_w)_j) = e_j,
\tag{12}
$$

*where $e_j = c_{ij}q_{ij} + c'_{ij}(1 - q_{ij}), \forall i$.*

Then we derive the probability that the watermarks decoded by the surrogate decoders are flipped when the watermark decoded by the target decoder is flipped after adding the perturbation as follows:

**Lemma 2.** *For any watermarked image $x_w$ and $\delta$ satisfying the constraints of the optimization problem in Equation 3, the jth bit of the watermark decoded by $S_i$ for the perturbed image is the inverse of the jth bit of the watermark decoded by $S_i$ for the watermarked image with probability $\frac{c_{ij}q_{ij}}{c_{ij}q_{ij} + c'_{ij}(1 - q_{ij})}$, given that the watermarks for the perturbed image and the watermarked image decoded by $T$ are inverse. Formally, for $j = 1, 2, \cdots, n$, we have the following:*

$$
\begin{aligned}
&Pr(S_i(x_w + \delta)_j = 1 - S_i(x_w)_j \mid T(x_w + \delta)_j = 1 - T(x_w)_j) \\
&= \frac{c_{ij}q_{ij}}{c_{ij}q_{ij} + c'_{ij}(1 - q_{ij})}.
\end{aligned}
\tag{13}
$$

Finally, we derive the probability that the watermark decoded by the target decoder is flipped after adding the perturbation to the watermarked image, given the watermarks decoded by the $m$ surrogate decoders for the perturbed image. Formally, we have the following:

**Theorem 3.** *For any watermarked image $x_w$ and $\delta$ satisfying the constraints of the optimization problem in Equation 3, the jth bit of the watermark decoded by $T$ for the perturbed image is the inverse of the jth bit of the watermark decoded by $T$ for the watermarked image with probability $p_j$, when the watermarks for the perturbed image decoded by the $m$ surrogate decoders are given. Formally, for $j = 1, 2, \cdots, n$, we have the following:*

$$
\begin{aligned}
&Pr(T(x_w + \delta)_j = 1 - T(x_w)_j \mid S_1(x_w + \delta)_j, \cdots, S_m(x_w + \delta)_j) \\
&= p_j,
\end{aligned}
\tag{14}
$$

*where $p_j = min(e_j \prod_{i \in M_{j1}} \frac{c_{ij}}{c_{ij}q_{ij} + c'_{ij}(1 - q_{ij})} \prod_{i \in M_{j2}} \frac{c'_{ij}}{c_{ij}q_{ij} + c'_{ij}(1 - q_{ij})}, 1)$, $M_{j1} = \{i \mid S_i(x_w + \delta)_j = 1 - S_i(x_w)_j\}$, and $M_{j2} = \{i \mid S_i(x_w + \delta)_j = S_i(x_w)_j\}$. $M_{j1}$ (or $M_{j2}$) is the set of surrogate decoders whose jth bits of the decoded watermarks flip (or not flip) after adding the perturbation to the watermarked image.*

### B.4 Deriving $Pr(T(x_w + \delta)_j = w_j)$

The bitwise accuracy between the decoded watermark $T(x_w + \delta)$ and the ground-truth watermark $w$ is used to detect whether the perturbed image $x_w + \delta$ is AI-generated. Therefore, given the watermarks decoded by the $m$ surrogate decoders for the perturbed image $x_w + \delta$, we derive an upper bound and a lower bound of the probability that the watermark decoded by the target decoder matches with the ground-truth watermark after adding the perturbation. Note that we consider $1 \leq j \leq n_t$ in this section. Formally, based on the theorems in Section B.3, we derive the Theorem 1 in Section 5.

Our theoretical analysis demonstrates that the probability of the $j$th bit of the watermark decoded by $T$ for the perturbed image matching with the $j$th bit of the ground-truth watermark can be bounded, which can be used to compute the upper bound and lower bound of the expected bitwise accuracy between $T(x_w + \delta)$ and $w$. If such upper bound and lower bound fall between $\tau$ and $1 - \tau$, the perturbed image is expected to evade the target watermark-based detector.

## C Proof of Lemma 1

Based on Definition 1, 2, and 3, we have the followings:

$$
\begin{aligned}
& Pr(T(x_w + \delta)_j = 1 - T(x_w)_j \mid S_i(x_w + \delta)_j = 1 - S_i(x_w)_j) \\
& = Pr(T(x_w + \delta)_j = 1 - T(x_w)_j, T(x_w)_j = S_i(x_w)_j \mid \\
& \quad S_i(x_w + \delta)_j = 1 - S_i(x_w)_j) \\
& \quad + Pr(T(x_w + \delta)_j = 1 - T(x_w)_j, T(x_w)_j = 1 - S_i(x_w)_j \mid \\
& \quad S_i(x_w + \delta)_j = 1 - S_i(x_w)_j) \\
& = Pr(T(x_w + \delta)_j = 1 - T(x_w)_j \mid T(x_w)_j = S_i(x_w)_j, \\
& \quad S_i(x_w + \delta)_j = 1 - S_i(x_w)_j) \\
& \quad \times Pr(T(x_w)_j = S_i(x_w)_j \mid S_i(x_w + \delta)_j = 1 - S_i(x_w)_j) \\
& \quad + Pr(T(x_w + \delta)_j = 1 - T(x_w)_j \mid T(x_w)_j = 1 - S_i(x_w)_j, \\
& \quad S_i(x_w + \delta)_j = 1 - S_i(x_w)_j) \\
& \quad \times Pr(T(x_w)_j = 1 - S_i(x_w)_j \mid S_i(x_w + \delta)_j = 1 - S_i(x_w)_j) \\
& = Pr(T(x_w + \delta)_j = 1 - T(x_w)_j \mid T(x_w)_j = S_i(x_w)_j, \\
& \quad S_i(x_w + \delta)_j = 1 - S_i(x_w)_j) \\
& \quad \times Pr(T(x_w)_j = S_i(x_w)_j \mid S_i(x_w + \delta)_j = 1 - S_i(x_w)_j) \\
& \quad + (1 - Pr(T(x_w + \delta)_j = T(x_w)_j \mid T(x_w)_j = 1 - S_i(x_w)_j, \\
& \quad S_i(x_w + \delta)_j = 1 - S_i(x_w)_j)) \\
& \quad \times (1 - Pr(T(x_w)_j = S_i(x_w)_j \mid S_i(x_w + \delta)_j = 1 - S_i(x_w)_j)) \\
& = Pr(T(x_w + \delta)_j = S_i(x_w + \delta)_j \mid T(x_w)_j = S_i(x_w)_j, \\
& \quad S_i(x_w + \delta)_j = 1 - S_i(x_w)_j) \\
& \quad \times Pr(T(x_w)_j = S_i(x_w)_j \mid S_i(x_w + \delta)_j = 1 - S_i(x_w)_j) \\
& \quad + (1 - Pr(T(x_w + \delta)_j = S_i(x_w + \delta)_j \mid \\
& \quad T(x_w)_j = 1 - S_i(x_w)_j, S_i(x_w + \delta)_j = 1 - S_i(x_w)_j)) \\
& \quad \times (1 - Pr(T(x_w)_j = S_i(x_w)_j \mid S_i(x_w + \delta)_j = 1 - S_i(x_w)_j)) \\
& = a_{ij} k_{ij} + (1 - a'_{ij})(1 - k_{ij}) \\
& = c_{ij}.
\end{aligned}
$$

Similarly, we have the following:

$$
\begin{aligned}
& Pr(T(x_w + \delta)_j = 1 - T(x_w)_j \mid S_i(x_w + \delta)_j = S_i(x_w)_j) \\
& = Pr(T(x_w + \delta)_j = 1 - T(x_w)_j, T(x_w)_j = S_i(x_w)_j \mid \\
& \quad S_i(x_w + \delta)_j = S_i(x_w)_j) + Pr(T(x_w + \delta)_j = 1 - T(x_w)_j, \\
& \quad T(x_w)_j = 1 - S_i(x_w)_j \mid S_i(x_w + \delta)_j = S_i(x_w)_j)
\end{aligned}
$$

$$\begin{aligned}
&= Pr(T(x_w + \delta)_j = 1 - T(x_w)_j \mid T(x_w)_j = S_i(x_w)_j, \\
&\quad S_i(x_w + \delta)_j = S_i(x_w)_j) \\
&\quad \times Pr(T(x_w)_j = S_i(x_w)_j \mid S_i(x_w + \delta)_j = S_i(x_w)_j) \\
&\quad + Pr(T(x_w + \delta)_j = 1 - T(x_w)_j \mid T(x_w)_j = 1 - S_i(x_w)_j, \\
&\quad S_i(x_w + \delta)_j = S_i(x_w)_j) \\
&\quad \times Pr(T(x_w)_j = 1 - S_i(x_w)_j \mid S_i(x_w + \delta)_j = S_i(x_w)_j) \\
&= (1 - Pr(T(x_w + \delta)_j = T(x_w)_j \mid T(x_w)_j = S_i(x_w)_j, \\
&\quad S_i(x_w + \delta)_j = S_i(x_w)_j)) \\
&\quad \times Pr(T(x_w)_j = S_i(x_w)_j \mid S_i(x_w + \delta)_j = S_i(x_w)_j) \\
&\quad + Pr(T(x_w + \delta)_j = 1 - T(x_w)_j \mid T(x_w)_j = 1 - S_i(x_w)_j, \\
&\quad S_i(x_w + \delta)_j = S_i(x_w)_j) \\
&\quad \times (1 - Pr(T(x_w)_j = S_i(x_w)_j \mid S_i(x_w + \delta)_j = S_i(x_w)_j)) \\
&= (1 - Pr(T(x_w + \delta)_j = S_i(x_w + \delta)_j \mid T(x_w)_j = S_i(x_w)_j, \\
&\quad S_i(x_w + \delta)_j = S_i(x_w)_j)) \\
&\quad \times Pr(T(x_w)_j = S_i(x_w)_j \mid S_i(x_w + \delta)_j = S_i(x_w)_j) \\
&\quad + Pr(T(x_w + \delta)_j = S_i(x_w + \delta)_j \mid T(x_w)_j = 1 - S_i(x_w)_j, \\
&\quad S_i(x_w + \delta)_j = S_i(x_w)_j) \\
&\quad \times (1 - Pr(T(x_w)_j = S_i(x_w)_j \mid S_i(x_w + \delta)_j = S_i(x_w)_j)) \\
&= (1 - b'_{ij})k'_{ij} + b_{ij}(1 - k'_{ij}) \\
&= c'_{ij}.
\end{aligned}$$

## D  PROOF OF THEOREM 2

Based on Lemma 1 and Definition 4, we have the following:

$$\begin{aligned}
&Pr(T(x_w + \delta)_j = 1 - T(x_w)_j) \\
&= Pr(T(x_w + \delta)_j = 1 - T(x_w)_j, S_i(x_w + \delta) = S_i(x_w)) \\
&\quad + Pr(T(x_w + \delta)_j = 1 - T(x_w)_j, S_i(x_w + \delta) = 1 - S_i(x_w)) \\
&= Pr(T(x_w + \delta)_j = 1 - T(x_w)_j \mid S_i(x_w + \delta) = S_i(x_w)) \\
&\quad \times Pr(S_i(x_w + \delta) = S_i(x_w)) \\
&\quad + Pr(T(x_w + \delta)_j = 1 - T(x_w)_j \mid S_i(x_w + \delta) = 1 - S_i(x_w)) \\
&\quad \times Pr(S_i(x_w + \delta) = 1 - S_i(x_w)) \\
&= c'_{ij}(1 - q_{ij}) + c_{ij}q_{ij} \\
&= e_j.
\end{aligned}$$

## E  PROOF OF LEMMA 2

$$\begin{aligned}
&Pr(S_i(x_w + \delta)_j = 1 - S_i(x_w)_j \mid T(x_w + \delta)_j = 1 - T(x_w)_j) \\
&= \frac{Pr(S_i(x_w + \delta)_j = 1 - S_i(x_w)_j, T(x_w + \delta)_j = 1 - T(x_w)_j)}{Pr(T(x_w + \delta)_j = 1 - T(x_w)_j)} \\
&= Pr(T(x_w + \delta)_j = 1 - T(x_w)_j \mid S_i(x_w + \delta)_j = 1 - S_i(x_w)_j) \\
&\quad \times \frac{Pr(S_i(x_w + \delta)_j = 1 - S_i(x_w)_j)}{Pr(T(x_w + \delta)_j = 1 - T(x_w)_j)}.
\end{aligned}$$

Based on Lemma 1 and Theorem 2, we have the following:

$$\begin{aligned}
&Pr(T(x_w + \delta)_j = 1 - T(x_w)_j \mid S_i(x_w + \delta)_j = 1 - S_i(x_w)_j) \\
&\quad \times \frac{Pr(S_i(x_w + \delta)_j = 1 - S_i(x_w)_j)}{Pr(T(x_w + \delta)_j = 1 - T(x_w)_j)}
\end{aligned}$$

$$= \frac{c_{ij}q_{ij}}{c_{ij}q_{ij} + c'_{ij}(1 - q_{ij})}.$$

# F    PROOF OF THEOREM 3

$$Pr(T(x_w + \delta)_j = 1 - T(x_w)_j \mid S_1(x_w + \delta)_j, \cdots, S_m(x_w + \delta)_j)$$
$$= Pr(S_1(x_w + \delta)_j, \cdots, S_m(x_w + \delta)_j \mid$$
$$T(x_w + \delta)_j = 1 - T(x_w)_j)$$
$$\times \frac{Pr(T(x_w + \delta)_j = 1 - T(x_w)_j)}{Pr(S_1(x_w + \delta)_j, \cdots, S_m(x_w + \delta)_j)}.$$

Based on Assumption 2 and 3, we have the following:

$$Pr(S_1(x_w + \delta)_j, \cdots, S_m(x_w + \delta)_j \mid T(x_w + \delta)_j = 1 - T(x_w)_j)$$
$$\times \frac{Pr(T(x_w + \delta)_j = 1 - T(x_w)_j)}{Pr(S_1(x_w + \delta)_j, \cdots, S_m(x_w + \delta)_j)}$$
$$= Pr(S_1(x_w + \delta)_j \mid T(x_w + \delta)_j = 1 - T(x_w)_j)$$
$$\cdots Pr(S_m(x_w + \delta)_j \mid T(x_w + \delta)_j = 1 - T(x_w)_j)$$
$$\times \frac{Pr(T(x_w + \delta)_j = 1 - T(x_w)_j)}{Pr(S_1(x_w + \delta)_j) \cdots Pr(S_m(x_w + \delta)_j)}. \tag{15}$$

Given that $M_{j1} = \{i \mid S_i(x_w + \delta)_j = 1 - S_i(x_w)_j\}$, we have the following according to Lemma 2:

$$Pr(S_i(x_w + \delta)_j \mid T(x_w + \delta)_j = 1 - T(x_w)_j)$$
$$= Pr(S_i(x_w + \delta)_j = 1 - S_i(x_w) \mid T(x_w + \delta)_j = 1 - T(x_w)_j)$$
$$= \frac{c_{ij}q_{ij}}{c_{ij}q_{ij} + c'_{ij}(1 - q_{ij})}, \forall i \in M_{j1}. \tag{16}$$

Given that $M_{j2} = \{i \mid S_i(x_w + \delta)_j = S_i(x_w)_j\}$, we have:

$$Pr(S_i(x_w + \delta)_j \mid T(x_w + \delta)_j = 1 - T(x_w)_j)$$
$$= 1 - Pr(S_i(x_w + \delta)_j = 1 - S_i(x_w) \mid$$
$$T(x_w + \delta)_j = 1 - T(x_w)_j)$$
$$= 1 - \frac{c_{ij}q_{ij}}{c_{ij}q_{ij} + c'_{ij}(1 - q_{ij})}$$
$$= \frac{c'_{ij}(1 - q_{ij})}{c_{ij}q_{ij} + c'_{ij}(1 - q_{ij})}, \forall i \in M_{j2}. \tag{17}$$

Then Equation 15 can be reformulated as follows:

$$Pr(S_1(x_w + \delta)_j \mid T(x_w + \delta)_j = 1 - T(x_w)_j)$$
$$\cdots Pr(S_m(x_w + \delta)_j \mid T(x_w + \delta)_j = 1 - T(x_w)_j)$$
$$\times \frac{Pr(T(x_w + \delta)_j = 1 - T(x_w)_j)}{Pr(S_1(x_w + \delta)_j) \cdots Pr(S_m(x_w + \delta)_j)}$$
$$= Pr(T(x_w + \delta)_j = 1 - T(x_w)_j)$$
$$\prod_{i \in M_{j1}} \frac{Pr(S_i(x_w + \delta)_j \mid T(x_w + \delta)_j = 1 - T(x_w)_j)}{Pr(S_i(x_w + \delta)_j)}$$
$$\prod_{i \in M_{j2}} \frac{Pr(S_i(x_w + \delta)_j \mid T(x_w + \delta)_j = 1 - T(x_w)_j)}{Pr(S_i(x_w + \delta)_j)}.$$

Then, based on Definition 4, Theorem 2, Equation 16, and Equation 17, we have the following:

$$Pr(T(x_w + \delta)_j = 1 - T(x_w)_j)$$

$$\prod_{i \in M_{j1}} \frac{Pr(S_i(x_w + \delta)_j \mid T(x_w + \delta)_j = 1 - T(x_w)_j)}{Pr(S_i(x_w + \delta)_j)}$$

$$\prod_{i \in M_{j2}} \frac{Pr(S_i(x_w + \delta)_j \mid T(x_w + \delta)_j = 1 - T(x_w)_j)}{Pr(S_i(x_w + \delta)_j)}$$

$$= e_j \prod_{i \in M_{j1}} \frac{c_{ij} q_{ij}}{c_{ij} q_{ij}^2 + c'_{ij}(1 - q_{ij}) q_{ij}}$$

$$\prod_{i \in M_{j2}} \frac{c'_{ij}(1 - q_{ij})}{c_{ij} q_{ij}(1 - q_{ij}) + c'_{ij}(1 - q_{ij})^2}$$

$$= min(e_j \prod_{i \in M_{j1}} \frac{c_{ij}}{c_{ij} q_{ij} + c'_{ij}(1 - q_{ij})}$$

$$\prod_{i \in M_{j2}} \frac{c'_{ij}}{c_{ij} q_{ij} + c'_{ij}(1 - q_{ij})}, 1)$$

$$= p_j.$$

## G    PROOF OF THEOREM 1

When $1 \leq j \leq n$, the conditional expectation of $|T(x_w + \delta)_j - T(x_w)_j|$ can be represented as:

$$E(|T(x_w + \delta)_j - T(x_w)_j| \mid S_1(x_w + \delta)_j, \cdots, S_m(x_w + \delta)_j)$$
$$= 0 \times Pr(T(x_w + \delta)_j = T(x_w)_j \mid$$
$$S_1(x_w + \delta)_j, \cdots, S_m(x_w + \delta)_j)$$
$$+ 1 \times Pr(T(x_w + \delta)_j = 1 - T(x_w)_j \mid$$
$$S_1(x_w + \delta)_j, \cdots, S_m(x_w + \delta)_j)$$
$$= Pr(T(x_w + \delta)_j = 1 - T(x_w)_j \mid$$
$$S_1(x_w + \delta)_j, \cdots, S_m(x_w + \delta)_j).$$

According to Theorem 3, we have:

$$Pr(T(x_w + \delta)_j = 1 - T(x_w)_j \mid S_1(x_w + \delta)_j, \cdots, S_m(x_w + \delta)_j)$$
$$= p_j.$$

According to Assumption 3, when $1 \leq j \leq n$, the conditional expectation of $|T(x_w)_j - w_j|$ can be represented as:

$$E(|T(x_w)_j - w_j| \mid S_1(x_w + \delta)_j, \cdots, S_m(x_w + \delta)_j)$$
$$= 0 \times Pr(T(x_w)_j = w_j \mid S_1(x_w + \delta)_j, \cdots, S_m(x_w + \delta)_j)$$
$$+ 1 \times Pr(T(x_w)_j = 1 - w_j \mid$$
$$S_1(x_w + \delta)_j, \cdots, S_m(x_w + \delta)_j)$$
$$= Pr(T(x_w)_j = 1 - w_j \mid S_1(x_w + \delta)_j, \cdots, S_m(x_w + \delta)_j)$$
$$= \frac{Pr(T(x_w)_j = 1 - w_j, S_1(x_w + \delta)_j, \cdots, S_m(x_w + \delta)_j)}{Pr(S_1(x_w + \delta)_j, \cdots, S_m(x_w + \delta)_j)}$$
$$= \frac{Pr(S_1(x_w + \delta)_j, \cdots, S_m(x_w + \delta)_j \mid T(x_w)_j = 1 - w_j)}{Pr(S_1(x_w + \delta)_j, \cdots, S_m(x_w + \delta)_j)}$$
$$\times Pr(T(x_w)_j = 1 - w_j)$$
$$= \frac{Pr(S_1(x_w + \delta)_j \mid T(x_w)_j = 1 - w_j)}{Pr(S_1(x_w + \delta)_j)}$$
$$\cdots \frac{Pr(S_m(x_w + \delta)_j \mid T(x_w)_j = 1 - w_j)}{Pr(S_m(x_w + \delta)_j)}$$

$$\times Pr(T(x_w)_j = 1 - w_j).$$

Since the flipping behavior of surrogate models is irrelevant to Definition 5, then we have the following:

$$E(|T(x_w)_j - w_j| \mid S_1(x_w + \delta)_j, \cdots, S_m(x_w + \delta)_j)$$
$$= Pr(T(x_w)_j = 1 - w_j).$$

Based on Definition 5, we have:

$$E(|T(x_w)_j - w_j| \mid S_1(x_w + \delta)_j, \cdots, S_m(x_w + \delta)_j)$$
$$= 1 - \beta_j.$$

Furthermore, the conditional expectation of $|T(x_w + \delta)_j - w_j|$ can be represented as:

$$E(|T(x_w + \delta)_j - w_j| \mid S_1(x_w + \delta)_j, \cdots, S_m(x_w + \delta)_j)$$
$$= 0 \times Pr(T(x_w + \delta)_j = w_j \mid S_1(x_w + \delta)_j, \cdots, S_m(x_w + \delta)_j)$$
$$+ 1 \times Pr(T(x_w + \delta)_j = 1 - w_j \mid$$
$$S_1(x_w + \delta)_j, \cdots, S_m(x_w + \delta)_j)$$
$$= Pr(T(x_w + \delta)_j = 1 - w_j \mid S_1(x_w + \delta)_j, \cdots, S_m(x_w + \delta)_j). \tag{18}$$

Based on the triangle inequality, we have the following:

$$E(|T(x_w + \delta)_j - w_j| \mid S_1(x_w + \delta)_j, \cdots, S_m(x_w + \delta)_j)$$
$$\leq min(E(|T(x_w + \delta)_j - T(x_w)_j| + |T(x_w)_j - w_j| \mid$$
$$S_1(x_w + \delta)_j, \cdots, S_m(x_w + \delta)_j), 1)$$
$$= min(1 - \beta_j + p_j, 1),$$

and the following:

$$E(|T(x_w + \delta)_j - w_j| \mid S_1(x_w + \delta)_j, \cdots, S_m(x_w + \delta)_j)$$
$$\geq E(||T(x_w + \delta)_j - T(x_w)_j| - |T(x_w)_j - w_j|| \mid$$
$$S_1(x_w + \delta)_j, \cdots, S_m(x_w + \delta)_j)$$
$$= |p_j + \beta_j - 1|.$$

Therefore, based on Equation 18, when $1 \leq j \leq n$, we have the following:

$$Pr(T(x_w + \delta)_j = w_j \mid S_1(x_w + \delta)_j, \cdots, S_m(x_w + \delta)_j)$$
$$= 1 - Pr(T(x_w + \delta)_j = 1 - w_j \mid$$
$$S_1(x_w + \delta)_j, \cdots, S_m(x_w + \delta)_j)$$
$$\geq 1 - min(1 - \beta_j + p_j, 1)$$
$$= max(\beta_j - p_j, 0),$$

and the following:

$$Pr(T(x_w + \delta)_j = w_j \mid S_1(x_w + \delta)_j, \cdots, S_m(x_w + \delta)_j)$$
$$\leq 1 - |p_j + \beta_j - 1|.$$

If $n_t > n$, when $n < j \leq n_t$, we have the following:

$$0 \leq Pr(T(x_w + \delta)_j = w_j \mid S_1(x_w + \delta)_j, \cdots, S_m(x_w + \delta)_j) \leq 1.$$

Therefore, we have the following:

$$Pr(T(x_w + \delta)_j = w_j \mid S_1(x_w + \delta)_j, \cdots, S_m(x_w + \delta)_j)$$
$$\geq \begin{cases} max(\beta_j - p_j, 0), & 1 \leq j \leq n, \\ 0, & n < j \leq n_t. \end{cases}$$
$$Pr(T(x_w + \delta)_j = w_j \mid S_1(x_w + \delta)_j, \cdots, S_m(x_w + \delta)_j)$$
$$\leq \begin{cases} 1 - |p_j + \beta_j - 1|, & 1 \leq j \leq n, \\ 1, & n < j \leq n_t. \end{cases}$$

## H  DETAILS OF DATASETS

For the target watermarking models, we use two public datasets (Wang et al., 2023; Turc & Nemade, 2022) generated by Stable Diffusion and Midjourney respectively. Following HiDDeN (Zhu et al., 2018), we randomly select 10,000 images from each dataset to train the target watermarking encoders and decoders. For testing, we randomly sample 1,000 images from the testing set of each dataset, embed the ground-truth watermark into each of them using a target encoder, and then find the perturbation to each watermarked image using different methods. To train the surrogate watermarking models, we sample 10,000 images from another public dataset (Images, 2023) generated by DALL-E 2, i.e., the surrogate dataset consists of these 10,000 images. The input image size of the watermarking models is $128 \times 128$ pixels.

## I  DETAILS OF TARGET WATERMARKING MODELS AND WATERMARK LENGTHS

For HiDDeN, we consider two architectures for the target watermarking model: CNN and ResNet. For the CNN architecture, the encoder consists of 4 convolutional blocks, while the decoder consists of 7 convolutional blocks. Each block integrates a Convolution layer, Batch Normalization, and ReLU activation. For the ResNet architecture, the encoder consists of 7 convolutional blocks and the decoder is the ResNet-18.

For StegaStamp, we use model architecture introduced by Tancik et al. (2020). For Stable Signature, we use the public watermarking model provided by Fernandez et al. (2023) as the target watermarking model. Smoothed HiDDeN and Smoothed StegaStamp are certifiably robust against bounded adversarial perturbations. For them, we adopt the regression smoothing (Jiang et al., 2024) to obtain the smoothed versions of the corresponding HiDDeN and StegaStamp watermarking models.

Additionally, we also employ different watermark lengths for the target watermarking models. Specifically, we evaluate our transfer attack on the target watermarking models with watermark lengths of 20 bits, 30 bits, and 64 bits. These variations include watermark lengths that are shorter than, equal to, and longer than those used by the surrogate watermarking models, offering a thorough analysis of our transfer attack's performance across different watermark lengths in the target watermarking model.

## J  DETAILS OF COMMON POST-PROCESSING METHODS

Specifically, we consider the following common post-processing methods.

**JPEG.** It is a commonly used compression method in digital imaging, which can reduce image file sizes while preserving a reasonable level of image quality. The quality of images processed through JPEG is governed by a quality factor $Q$. As the quality factor decreases, detecting the watermark in the post-processed images becomes more challenging, although this also results in a lower image quality.

**Gaussian noise.** This method involves adding statistical noise that follows a Gaussian distribution with a zero mean and a standard deviation of $\sigma$, which effectively simulates various environmental noise effects encountered in real-world scenarios. A larger $\sigma$ value leads to increased difficulty in watermark detection, but also results in lower image quality.

**Gaussian blur.** This method smooths the image by averaging pixel values with their neighbors. It applies a Gaussian filter with kernel size of $k \times k$ to an image, characterized by a bell-shaped curve with a zero mean and a standard deviation of $\sigma$. A larger $\sigma$ causes more pronounced blurring, which results in lower watermark detection rate and image quality. Following Jiang et al. (2023), we set $k = 5$ and vary $\sigma$.

**Brightness/Contrast.** This method modifies the brightness and contrast of an image by adjusting pixel values throughout the image. Specifically, it operates by either increasing or decreasing these values to make the image brighter or darker. The process is regulated by two factors: a brightness factor $B$ and a contrast factor $C$. Formally, for each pixel value $v$, the method transforms it to $Cv + B$. Following Jiang et al. (2023), we set $B = 0.2$ and vary $C$.

# K  DETAILS OF TRANSFER ATTACKS

For the former 5 transfer attacks, WEvade-B-S is watermark-based, while the other four are classifier-based. In particular, MI-CWA leverages the state-of-the-art multiple surrogate classifiers based transfer adversarial examples.

**WEvade-B-S (Jiang et al., 2023).** This method trains one surrogate watermarking model. A watermarked image is perturbed such that the watermark decoded by the surrogate decoder for the perturbed image matches with a preset random watermark. In our experiments, we train the surrogate watermarking model on 10,000 images generated by DALL-E 2. To give advantages to this transfer attack, we assume the attacker knows the architecture of the target watermarking model and uses it for the surrogate watermarking model.

**AdvEmb-RN18 (An et al., 2024).** This method uses a ResNet-18 pretrained on ImageNet to generate a feature embedding for a watermarked image. Then it perturbs the image such that its embedding lies far from the one of the watermarked image.

**AdvCls-Real&WM (An et al., 2024).** This method trains a surrogate classifier using watermarked and non-watermarked images. The watermarked images are generated by the target GenAI and are watermarked by the target watermarking model, and the non-watermarked images are drawn from a distribution different from the one of the images generated by the target GenAI. A watermarked image is perturbed such that it is misclassified by the surrogate classifier as a non-watermarked image. In our experiments, we utilize images generated by DALL-E 2 as the non-watermarked images. More specifically, the training set for the surrogate classifier comprises 8,000 images generated by Stable Diffusion (or Midjourney) and watermarked by the target watermarking model, and 8,000 non-watermarked images generated by DALL-E 2. The surrogate classifier is based on the ResNet-18 architecture.

**AdvCls-Enc-WM1&WM2.** This method is the same as AdvCls-Real&WM except for the training data used for the surrogate classifier. It assumes that the attacker has access to the target watermarking encoder and can use it to embed any watermark into an image. The surrogate classifier is trained to distinguish the images embedded with the ground-truth watermark and those embedded with an attacker-chosen watermark. In our experiments, the training set for the surrogate classifier comprises 8,000 images generated by the target GenAI and watermarked by the target watermarking model with one watermark, and 8,000 images generated by DALL-E 2 and watermarked by the target watermarking model with another watermark, where both watermarks are randomly picked.

**MI-CWA (Chen et al., 2024).** Existing classifier-based transfer attacks (i.e., the above three) only leverage one surrogate classifier. We extend state-of-the-art multiple surrogate classifiers based transfer adversarial examples (Chen et al., 2024) to watermarks. Given the same training dataset as AdvCls-Real&WM, we train 100 surrogate classifiers, each of which has the ResNet-18 architecture. Given a watermarked image, this transfer attack perturbs it such that it is misclassified by the surrogate classifiers as non-watermarked.

**DiffPure (Nie et al., 2022).** This method uses diffusion model to purify adversarial examples. We extend it to remove watermark. Specifically, DiffPure first adds Gaussian noise gradually to turn a watermarked image into a noised image. Then diffusion model is used to predict the noise step by step to get the denoised image. We use its public code.

# L  DETAILS OF PARAMETER SETTINGS

Specifically, for common post-processing methods, we consider the following range of parameters during adversarial training for a target watermarking model: $Q \in [10, 99]$ for JPEG, $\sigma \in [0, 0.1]$ for Gaussian noise, $\sigma \in [0, 2.0]$ for Gaussian blur, and $C \in [1, 3]$ for Brightness/Contrast. For surrogate watermarking models, we adopt a smaller range of parameters for some common post-processing methods during adversarial training to achieve weaker robustness than the target watermarking model as follows: $Q \in [50, 99]$ for JPEG, $\sigma \in [0, 0.1]$ for Gaussian noise, $\sigma \in [0, 1.0]$ for Gaussian blur, and $C \in [1, 3]$ for Brightness/Contrast.

By default, we set maximum number of iterations $max\_iter = 5,000$, perturbation budget $r = 0.25$, sensitivity $\epsilon = 0.2$, and learning rate $\alpha = 0.1$ for our transfer attack. Unless otherwise men-

tioned, we use Inverse-Decode to select a target watermark for a surrogate decoder, and Ensemble-Optimization to find the perturbation. $\alpha$ is increased when the number of surrogate watermarking models increases in order to satisfy the constraints of our optimization problem within $5,000$ iterations. The detailed settings for $\alpha$ for different number of surrogate models are shown in Table 1 in Appendix. Moreover, we use $\ell_2$-distance as the distance metric $l(\cdot, \cdot)$ for two watermarks.

For the detection threshold $\tau$, we set it based on the watermark length of the target watermarking model. Specifically, we set $\tau$ to be a value such that the false positive rate of the watermark-based detector is no larger than $10^{-4}$ when the double-tail detector is employed. Specifically, $\tau$ is set to be 0.9, 0.83, and 0.73 for the target watermarking models with watermark lengths of 20 bits, 30 bits, and 64 bits, respectively.

