# OpenReview forum: "A Transfer Attack to Image Watermarks"
_ICLR.cc/2025/Conference — ICLR 2025 Poster_

### Official Review · Reviewer_F4Nf · 2024-10-28

**Soundness:** 4
**Presentation:** 3
**Contribution:** 3
**Rating:** 8
**Confidence:** 4

**Summary:**

This paper develops an attack that evades a target image watermarking detector, without acquiring the access to this detector's model or API. The proposed method adds a perturbation trained to mislead multiple surrogate detector models, and hopes that this perturbation can transfer to the target detector and therefore succeed. The experiment results on various datasets demonstrate the effectiveness of the proposed method, which outperforms many existing attacks and post-processing methods.

**Strengths:**

1. The paper is well-written and easy to follow.
2. This paper proposed an intuitive yet attractive method that seems work very well in the experimental studies.
3.  The experimental studies are comprehensive.

**Weaknesses:**

**Major Concerns:**

1. Theoretical contributions. While authors provide some theoretical analysis, I think the current results are not sufficient to be claimed as signficant. The reasons are as follows:
(1) It is in doubt whether those independence assumptions will hold. As $\delta$ is obtained by solving Eq. (3), which optimizes the chances that $m$ decoders will extract carefully designed watermarks from an image perturbed by $\delta$, it is likely that coordinates of decoders' output is correlated. The authors may want to provide additional envidence to validate their assumptions.
(2) The analysis only applies to inverse-decode.
(3) The bound given by the theorems are not available in practice. Recall that this paper considers the setting where $T$ and its output is unknown to the attacker. In the meanwhile, the parameters including $a, b,\beta, p$ involves the unknown target decoder $T$ and unknown data distribution.

2. Empirical results. The empirical results are impressive, and it raises my question why it significantly outperforms SOTA methods? Could the author provide any justification and insight for the superior performance of the proposed method?


**Minor Concerns:**

1. Since the proposed method has to train a bunch of surrogate models, will it be computationally expensive? Also, every time this algorithm has to solve Eq. (3), which may also be expensive. The authors may want to add a table comparing the FLOPs and time used by different methods.

**Questions:**

Please see the weakness section above.

**Details Of Ethics Concerns:**

The proposed method might be used to compromise the identity and authenticity of AI-generated images. Therefore, a discussion of potential negative societal influence and possible mitigation approaches is suggested to be included.

---

> ### Author Response · Authors · 2024-11-20
> **Response to Reviewer F4Nf**
>
> We sincerely appreciate your constructive comments and insightful suggestions. Please find our responses below.
>
> > **Weakness 1**:  While authors provide some theoretical analysis, I think the current results are not sufficient to be claimed as significant. The reasons are as follows: (1) It is in doubt whether those independence assumptions will hold. As $\delta$ is obtained by solving Eq. (3), which optimizes the chances that $m$ decoders will extract carefully designed watermarks from an image perturbed by $\delta$, it is likely that coordinates of decoders' output are correlated. The authors may want to provide additional evidence to validate their assumptions.
>
> **Response**: Thank you for your insightful suggestions. To support this assumption, we tested the independence of the watermarks decoded by surrogate models from perturbed images generated by our attack. Specifically, we randomly sampled 100 pairs of surrogate watermarking models from the 100 surrogate models in our experiments. For each bit of the watermark and pair of surrogate models, we performed the Chi-square independence test with a confidence level of 0.95 using the decoded watermarks from the 1,000 perturbed images. Then we calculated the percentage of the 100 pairs of the surrogate models that passed the independence test for each bit.
>
> | **Bit** | **Percentage** | **Bit** | **Percentage** | **Bit** | **Percentage** |
> |---------|----------------|---------|----------------|---------|----------------|
> | 1       | 0.86           | 11      | 0.88           | 21      | 0.90           |
> | 2       | 0.93           | 12      | 0.91           | 22      | 0.87           |
> | 3       | 0.89           | 13      | 0.91           | 23      | 0.87           |
> | 4       | 0.89           | 14      | 0.87           | 24      | 0.91           |
> | 5       | 0.90           | 15      | 0.88           | 25      | 0.90           |
> | 6       | 0.90           | 16      | 0.89           | 26      | 0.92           |
> | 7       | 0.89           | 17      | 0.92           | 27      | 0.91           |
> | 8       | 0.85           | 18      | 0.92           | 28      | 0.89           |
> | 9       | 0.88           | 19      | 0.88           | 29      | 0.88           |
> | 10      | 0.92           | 20      | 0.92           | 30      | 0.90           |
>
> The results, shown in the above table and Figure 14 in the Appendix of our revised paper, reveal that nearly 90% of pairs are independent for each bit. This finding provides strong evidence supporting our assumption.
>
> > **Weakness 2**: The analysis only applies to inverse-decode.
>
> **Response**: Thank you for your valuable comment. We would like to mention that our theoretical analysis can be straightforwardly extended to Random-Different, as all the assumptions used to derive the bounds for Inverse-Decode, particularly the independence assumption, still hold for Random-Different. Given the randomness of watermark selection, Random-Different can effectively be viewed as the roughly 50% inverse version of Inverse-Decode. If needed, we can provide further explanations and a detailed theoretical analysis for Random-Different.
>
> > **Weakness 3**: The bounds given by the theorems are not available in practice. Recall that this paper considers the setting where $T$ and its output is unknown to the attacker. In the meanwhile, the parameters including $a$, $b$, $\beta$, $p$ involve the unknown target decoder $T$ and unknown data distribution.
>
> **Response**: Thank you for your valuable comment. We acknowledge that calculating our theoretical bounds requires more information than attackers typically possess. However, the primary purpose of our theoretical results is to provide insights into why our transfer attack succeeds and how effectively it can perform. These bounds are not meant to be directly usable by attackers but rather serve to analyze the underlying mechanisms and potential of our approach. Thus, the need for additional knowledge is acceptable in the context of theoretical analysis.

---

> ### Author Response · Authors · 2024-11-20
> **Response to Reviewer F4Nf**
>
> > **Weakness 4**: The empirical results are impressive, and it raises my question why it significantly outperforms SOTA methods? Could the author provide any justification and insight for the superior performance of the proposed method?
>
> **Response**: Thank you for your valuable question. Most other transfer-based attacks simply apply adversarial examples to the watermark and craft perturbations based on classifiers, without accounting for the unique properties of watermarking. Consequently, these methods achieve limited transferability. A more detailed discussion can be found in Section 2.2 in our paper.
> For DiffPure, it utilizes a diffusion model to purify the image by removing the watermark. However, the diffusion process significantly alters the watermarked image by introducing large perturbations. As a result, DiffPure achieves limited success when the $\ell_{\infty}$ norm of the perturbation or the SSIM between the perturbed and watermarked image is constrained, as demonstrated in Figure 11 of our revised paper. In contrast, we observe that our transfer attack consistently outperforms all baseline methods in terms of evasion rate while maintaining comparable image quality (SSIM).
>
> > **Weakness 5**: Since the proposed method has to train a bunch of surrogate models, will it be computationally expensive? Also, every time this algorithm has to solve Eq. (3), which may also be expensive. The authors may want to add a table comparing the FLOPs and time used by different methods.
>
> **Response**: Thank you for your helpful suggestions. To analyze the computational cost of our transfer attack and compare it with existing transfer attacks, we have added a table summarizing the training and inference times for our attack and six baseline methods used in our experiments, all evaluated on the same device.
>
> | **Method**               | **Training Time (h)**            | **Inference Time per Image (s)** |
> |--------------------------|-----------------------------------|-----------------------------------|
> | AdvEmb-RN18              | $\Delta_{\text{RN18}}$           | 72.09                            |
> | AdvCls-Real&WM           | $\Delta_{\text{RN18}}$ + 0.11    | 77.04                            |
> | AdvCls-Enc-WM1&WM2       | $\Delta_{\text{RN18}}$ + 0.11    | 77.70                            |
> | MI-CWA                   | $\Delta_{\text{RN18}}$ + 0.11    | 27.01                            |
> | DiffPure                 | $\Delta_{\text{Diff}}$           | 8.96                             |
> | WEvade-B-S               | 17.06                            | 187.03                           |
> | Ours                     | 6.50                             | 177.97                           |
>
> Training time refers to the duration required to train a surrogate model, while inference time denotes the time needed to optimize the perturbation for an image. All measurements were conducted on a single NVIDIA RTX-6000 GPU. Specifically, $\Delta_{\text{RN18}}$ represents the pre-training time for ResNet-18~\cite{} on ImageNet, and $\Delta_{\text{Diff}}$ corresponds to the training time for the unconditional diffusion model used by DiffPure. The target watermarking model in our experiments is ResNet, and all perturbations were optimized under the constraint $SSIM \geq 0.9$. The evasion rates for each method are provided in the corresponding data points in Figure 11 in the Appendix.
>
> The results indicate that our attack requires an acceptable training time, as the training of multiple surrogate models can be parallelized. Additionally, the trained surrogate models can be reused to attack multiple target watermarking models, further improving overall efficiency. While the inference time of our attack exceeds that of five out of six baseline methods, it remains feasible for attackers with sufficient computational resources. Notably, the optimization of $\delta$ in Eq. (3) for different watermarked images can be parallelized, ensuring that the overall computational cost remains reasonable. This discussion has been incorporated into Section 7 in our revised paper.

---

> > ### Comment · Reviewer_F4Nf · 2024-11-21
> >
> > I thank the authors for their detailed responses and additional experiments to address my concerns. I raise my score to 8.

---

> > > ### Author Response · Authors · 2024-11-21
> > >
> > > Thank you again for your constructive comments and raising the score!

---

### Official Review · Reviewer_xUbX · 2024-11-03

**Soundness:** 3
**Presentation:** 2
**Contribution:** 2
**Rating:** 6
**Confidence:** 4

**Summary:**

This paper demonstrates that watermark-based detection methods for AI-generated images are vulnerable to transfer attacks in a no-box setting. Specifically, an attacker can effectively remove the watermark from a given watermarked image by applying a perturbation, which can be identified through ensembling multiple surrogate watermarking models.

**Strengths:**

This paper presents a practical attack effective across multiple watermarking methods
The issue of watermarking the outputs of generative models is timely and interesting.

**Weaknesses:**

To ensure high image quality, the L infinity bound on perturbations should be kept low, as higher values (e.g., 0.2 as shown in Figures 3 and 4) can noticeably degrade image quality, reducing practicality. For a more realistic assessment, the paper should include comparisons using a set of small enough  L infinity bound (such as 0.03, 0.05) bounds to demonstrate the proposed method’s effectiveness under conditions that minimize visible impact.
If Section 5 is one of the paper’s main contributions, it would be better placed in the main text rather than in the appendix.

**Questions:**

See weakness

---

> ### Author Response · Authors · 2024-11-20
> **Response to Reviewer xUbX**
>
> We sincerely appreciate your constructive comments and insightful suggestions. Please find our responses below.
>
> > **Weakness 1**: To ensure high image quality, the L infinity bound on perturbations should be kept low, as higher values (e.g., 0.2 as shown in Figures 3 and 4) can noticeably degrade image quality, reducing practicality. For a more realistic assessment, the paper should include comparisons using a set of small enough L infinity bound (such as 0.03, 0.05) bounds to demonstrate the proposed method’s effectiveness under conditions that minimize visible impact.
>
> **Response**: Thank you for highlighting this. We did new experiments based on the suggestion. The below table and Figure 11 in the Appendix of our revised paper presents the evasion rates of all attacks across a wide range of $SSIM$ constraints.
> | **Method**               | **1.00** | **0.99** | **0.95** | **0.90** | **0.85** | **0.80** | **0.75** | **0.70** | **0.65** | **0.60** |
> |---------------------------|----------|----------|----------|----------|----------|----------|----------|----------|----------|----------|
> | AdvEmb-RN18              | 0.00     | 0.00     | 0.00     | 0.00     | 0.00     | 0.01     | 0.01     | 0.02     | 0.06     | 0.12     |
> | AdvCls-Real&WM           | 0.00     | 0.00     | 0.00     | 0.00     | 0.00     | 0.00     | 0.00     | 0.00     | 0.00     | 0.00     |
> | AdvCls-Enc-WM1&WM2       | 0.00     | 0.04     | 0.10     | 0.10     | 0.10     | 0.10     | 0.10     | 0.10     | 0.10     | 0.10     |
> | MI-CWA                   | 0.00     | 0.00     | 0.00     | 0.03     | 0.07     | 0.12     | 0.20     | 0.34     | 0.64     | 0.71     |
> | DiffPure                 | 0.00     | 0.00     | 0.00     | 0.04     | 0.19     | 0.33     | 0.73     | 1.00     | 1.00     | 1.00     |
> | WEvade-B-S               | 0.00     | 0.002    | 0.002    | 0.002    | 0.002    | 0.002    | 0.002    | 0.002    | 0.002    | 0.002    |
> | Ours                     | 0.00     | 0.218    | 0.997    | 1.00     | 1.00     | 1.00     | 1.00     | 1.00     | 1.00     | 1.00     |
>
> Our results show that our transfer attack consistently outperforms all baseline attacks. This highlights the effectiveness of our attack compared to existing ones.
>
> In this table, we focus on SSIM rather than the $\ell_{\infty}$ norm because SSIM provides a perceptually meaningful evaluation of image quality by accounting for structural similarity rather than isolated pixel differences. While high $\ell_{\infty}$ values in our method are typically localized to a small number of pixels with minimal perceptual impact, SSIM better reflects the overall quality and visual coherence of the image, making it a more suitable metric for this comparison.
>
> > **Weakness 2**: If Section 5 is one of the paper’s main contributions, it would be better placed in the main text rather than in the appendix.
>
> **Response**: Thank you for your valuable and helpful suggestions. In the revised version of our paper, we have moved the main theorem, which establishes the upper and lower bounds on the transferability of our transfer attack—our primary theoretical contribution—back to the main body.

---

> ### Author Response · Authors · 2024-11-22
>
> Dear Reviewer,
>
> Thank you once again for your constructive comments and valuable feedback! We have thoroughly addressed your suggestions by conducting new experiments and revising our paper. Could you please kindly review our responses and let us know if you have any further comments or suggestions?
>
> We would greatly appreciate it if you could reconsider your score in light of our new experimental results and the improvements made in the paper.
>
> Thank you for your time and consideration!

---

> ### Author Response · Authors · 2024-11-25
>
> Dear Reviewer,
>
> We wanted to kindly remind you about the responses and updates we shared regarding your constructive comments and valuable feedback on our paper. We have addressed your suggestions by conducting new experiments and revising the manuscript accordingly.
>
> Could you please review our responses at your earliest convenience? If you have any further comments or suggestions, we would be more than happy to address them. Additionally, we kindly ask you to reconsider your score in light of the new experimental results and the improvements made in the paper.
>
> As the discussion phase deadline is approaching, we greatly appreciate your time and attention to this matter.
>
> Thank you again for your thoughtful feedback and for considering our revised work.

---

> > ### Comment · Reviewer_xUbX · 2024-11-25
> > **Official Comment by xUbX**
> >
> > Thank you for your response and the additional experiments. My main concern was the degradation of image quality while achieving a high evasion rate. Your experiments demonstrate an evasion rate of 0.997 under an SSIM constraint of 0.95, which addresses much of my concern. Based on this, I am willing to increase my score to 6.

---

> > > ### Author Response · Authors · 2024-11-26
> > >
> > > Thank you again for your valuable comments and raising the score!

---

> ### Author Response · Authors · 2024-11-25
>
> Thank you again for your constructive comments and promise to raise the score!

---

> ### Author Response · Authors · 2024-11-26
>
> Dear Reviewer xUbX,
>
> Thank you once again for your review and constructive feedback. Your pre-rebuttal rating was 5. We’re delighted to know that you found our rebuttal and new experiments satisfactory, and we appreciate your kind promise to raise your rating to be 6. As a gentle reminder, we kindly ask that you update your rating at your convenience to ensure your post-rebuttal evaluation is accurately reflected in the system. Thank you again for your time and support!

---

### Official Review · Reviewer_k3og · 2024-11-03

**Soundness:** 3
**Presentation:** 4
**Contribution:** 2
**Rating:** 6
**Confidence:** 3

**Summary:**

This paper proposed an evasion attack to interfere with the detection of watermarked images under the no-box setting, i.e. the attacker has no access to the watermark detector. They approach the objective through the transfer attack, where they train surrogate watermark models and find adversarial perturbation to fool the detection of local surrogate models. Both theoretical analysis and empirical evidence demonstrate that this transfer attack method can successfully undermine the detection capabilities of learning-based watermark systems across various watermarking scenarios.

**Strengths:**

1. This paper is well-written with clear motivation. The no-box setting is practical.
2. The experiments and theoretical results support their claims and method design.
3. The transfer results are effective against several learning-based watermarking systems.

**Weaknesses:**

1. While I acknowledge the authors have assumed that the attackers should have sufficient computational resources, the main concern of this paper is the computational cost. The costs associated with both surrogate models and perturbation discovery are large which prevents the practical usage of this method.
2. For the transfer attack to a specific architecture such as ResNet-18, did you use a similar architecture in your surrogate models, for instance, ResNet-X? Besides, have you considered attacks with totally different architectures, such as transformer-based watermark models?
3. Another weakness of this paper is that the transfer attack primarily focuses on learning-based watermarks, neglecting non-learning-based watermark variants that may exhibit greater robustness than the baselines discussed here, such as variants of Tree-Ring [1][2], etc. Besides, can the proposed method attack the 0-bit watermark?
The cost of attacks escalates significantly with complex architectures a
4. The attack cost is significantly increased with complex architectures and longer watermarks. This poses a hindrance to the practical implementation of transfer attacks involving high-capacity watermarks [3].

[1] RingID: Rethinking Tree-Ring Watermarking for Enhanced Multi-Key Identification

[2] Gaussian Shading: Provable Performance-Lossless Image Watermarking for Diffusion Models

[3] A Robust Image Watermarking System Based on Deep Neural Networks

**Questions:**

Please see the weaknesses.

---

> ### Author Response · Authors · 2024-11-20
> **Response to Reviewer k3og**
>
> We sincerely appreciate your constructive comments and insightful suggestions. Please find our responses below.
>
> > **Weakness 1**: While I acknowledge the authors have assumed that the attackers should have sufficient computational resources, the main concern of this paper is the computational cost. The costs associated with both surrogate models and perturbation discovery are large which prevents the practical usage of this method.
>
> **Response**: Thank you for your insightful suggestions regarding the analysis of computational cost. To evaluate the computational cost  of our attack, we include the following table summarizing the training and inference times for our attack and six baseline methods used in our experiments.
>
> | **Method**               | **Training Time (h)**            | **Inference Time per Image (s)** |
> |--------------------------|-----------------------------------|-----------------------------------|
> | AdvEmb-RN18              | $\Delta_{\text{RN18}}$           | 72.09                            |
> | AdvCls-Real&WM           | $\Delta_{\text{RN18}}$ + 0.11    | 77.04                            |
> | AdvCls-Enc-WM1&WM2       | $\Delta_{\text{RN18}}$ + 0.11    | 77.70                            |
> | MI-CWA                   | $\Delta_{\text{RN18}}$ + 0.11    | 27.01                            |
> | DiffPure                 | $\Delta_{\text{Diff}}$           | 8.96                             |
> | WEvade-B-S               | 17.06                            | 187.03                           |
> | Ours                     | 6.50                             | 177.97                           |
>
> Training time refers to the duration required to train a surrogate model, while inference time denotes the time needed to optimize the perturbation for an image. All measurements were performed on a single NVIDIA RTX-6000 GPU. Specifically, $\Delta_{\text{RN18}}$ represents the pre-training time of ResNet-18 on ImageNet, and $\Delta_{\text{Diff}}$ corresponds to the training time of the unconditional diffusion model employed by DiffPure. The target watermarking model in our experiments is a ResNet, and all perturbations were optimized under the constraint $SSIM \geq 0.9$. The evasion rates for each method are presented in Figure 11 in the Appendix of our revised paper.
>
> For our method, the training time is 6.5 hours. Although extending this to multiple surrogate models increases the total duration, these models can be trained in parallel across GPUs and reused for various target watermarking models, which keeps the overall overhead manageable. While our inference time exceeds that of five out of six baseline methods, it remains feasible for attackers with adequate computational resources. This analysis has been included in Section 7 of our revised paper.
>
> > **Weakness 2**: For the transfer attack to a specific architecture such as ResNet-18, did you use a similar architecture in your surrogate models, for instance, ResNet-X? Besides, have you considered attacks with totally different architectures, such as transformer-based watermark models?
>
> **Response**: Thank you for raising this valuable question. For our surrogate models, we used a vanilla CNN with seven convolutional layers, which is entirely different from ResNet-18, as detailed in our Appendix. We have also considered watermarking models with totally different architectures. For instance, the StegaStamp method in our experiments employs a spatial transformer network, while the Stable Signature method utilizes a variational autoencoder (VAE) within the diffusion model for watermarking.
> Moreover, in Section 7, we also applied our transfer attack to DWT-DCT, a non-learning-based watermarking method. DWT-DCT has a qualitatively different architecture. Specifically, unlike the learning-based approaches examined in our experiments, DWT-DCT is a non-learning-based technique used by Stable Diffusion. This method employs the Discrete Wavelet Transform (DWT) to decompose an image into frequency sub-bands, applies the Discrete Cosine Transform (DCT) to blocks within selected sub-bands, and embeds the watermark by modifying specific frequency coefficients. The watermarked image is then reconstructed using inverse transforms.

---

> ### Author Response · Authors · 2024-11-20
> **Response to Reviewer k3og**
>
> > **Weakness 3**: Another weakness of this paper is that the transfer attack primarily focuses on learning-based watermarks, neglecting non-learning-based watermark variants that may exhibit greater robustness than the baselines discussed here, such as variants of Tree-Ring [1][2], etc. Besides, can the proposed method attack the 0-bit watermark?
>
> **Response**: Thank you for pointing out this. To include the Tree-Ring variants [1][2], we evaluated their robustness against common perturbations and presented the results in the below table (comparing evasion rates of non-learning-based watermarking models and HiDDeN under Gaussian noise with varying standard deviations) and Figure 6(c) in our revised paper.
>
> | **Method**          | **0.00** | **0.04** | **0.08** | **0.12** | **0.16** | **0.20** |
> |----------------------|----------|----------|----------|----------|----------|----------|
> | Tree-Ring           | 0.00     | 0.04     | 0.19     | 0.41     | 0.53     | 0.65     |
> | RingID              | 0.00     | 0.87     | 1.00     | 1.00     | 1.00     | 1.00     |
> | Gaussian Shading    | 0.00     | 0.005    | 0.03     | 0.09     | 0.32     | 0.46     |
> | HiDDeN             | 0.00     | 0.00     | 0.00     | 0.00     | 0.01     | 0.01     |
>
> Our findings demonstrate that non-learning-based methods are less robust compared to the learning-based methods we tested. Specifically, non-learning-based methods exhibit a noticeable rise in evasion rate even with relatively small perturbations (consistent with prior works). In contrast, learning-based methods maintain a low evasion rate under the same conditions.
> For this reason, our study primarily focuses on learning-based watermarks. However, we have also added a discussion of non-learning-based watermarks in Section 7 of our paper.
>
> [1] RingID: Rethinking Tree-Ring Watermarking for Enhanced Multi-Key Identification
>
> [2] Gaussian Shading: Provable Performance-Lossless Image Watermarking for Diffusion Models
>
> > **Weakness 4**: The attack cost is significantly increased with complex architectures and longer watermarks. This poses a hindrance to the practical implementation of transfer attacks involving high-capacity watermarks [3].
>
> **Response**: Thank you for pointing out this. To address it, we have added a table comparing the attack cost of our method with that of other baseline attacks.
>
> | **Method**               | **Training Time (h)**            | **Inference Time per Image (s)** |
> |--------------------------|-----------------------------------|-----------------------------------|
> | AdvEmb-RN18              | $\Delta_{\text{RN18}}$           | 72.09                            |
> | AdvCls-Real&WM           | $\Delta_{\text{RN18}}$ + 0.11    | 77.04                            |
> | AdvCls-Enc-WM1&WM2       | $\Delta_{\text{RN18}}$ + 0.11    | 77.70                            |
> | MI-CWA                   | $\Delta_{\text{RN18}}$ + 0.11    | 27.01                            |
> | DiffPure                 | $\Delta_{\text{Diff}}$           | 8.96                             |
> | WEvade-B-S               | 17.06                            | 187.03                           |
> | Ours                     | 6.50                             | 177.97                           |
>
> The results indicate that our attack requires an acceptable training time, as the training of multiple surrogate models can be parallelized. Additionally, the surrogate models only need to be trained once. While the inference time of our attack is longer than that of five out of six baseline methods, it remains feasible for attackers with adequate computational resources. We also discussed the implications of the high-capacity watermarks [3] on computational cost in Section 7 of our revised paper.
>
> [3] A Robust Image Watermarking System Based on Deep Neural Networks

---

> ### Author Response · Authors · 2024-11-22
>
> Dear Reviewer,
>
> Thank you once again for your constructive comments and valuable feedback! We have thoroughly addressed your suggestions by conducting new experiments and revising our paper. Could you please kindly review our responses and let us know if you have any further comments or suggestions?
>
> We would greatly appreciate it if you could reconsider your score in light of our new experimental results and the improvements made in the paper.
>
> Thank you for your time and consideration!

---

> ### Author Response · Authors · 2024-11-25
>
> Dear Reviewer,
>
> We wanted to kindly remind you about the responses and updates we shared regarding your constructive comments and valuable feedback on our paper. We have addressed your suggestions by conducting new experiments and revising the manuscript accordingly.
>
> Could you please review our responses at your earliest convenience? If you have any further comments or suggestions, we would be more than happy to address them. Additionally, we kindly ask you to reconsider your score in light of the new experimental results and the improvements made in the paper.
>
> As the discussion phase deadline is approaching, we greatly appreciate your time and attention to this matter.
>
> Thank you again for your thoughtful feedback and for considering our revised work.

---

> > ### Comment · Reviewer_k3og · 2024-11-25
> >
> > Thank you for your detailed response. After thoroughly reading your response, I have no further questions regarding this paper. In my view, the primary idea of this paper resembles using the model-ensemble to enhance adversarial transferability which has been extensively researched. Another point is that I still believe this method requires high computational resources, which is higher than that required for adversarial countermeasures. Given all, I would like to maintain my initial rating, or more accurately, rating it 7 would be appropriate. I am positive about this paper and lean towards acceptance as a poster.

---

> > > ### Author Response · Authors · 2024-11-25
> > > **Response to Reviewer k3og**
> > >
> > > Thank you for your thoughtful comments and for recommending our paper for acceptance. We greatly appreciate your positive feedback and your intent to assign a score of 7.
> > >
> > > > In my view, the primary idea of this paper resembles using the model-ensemble to enhance adversarial transferability which has been extensively researched.
> > >
> > > **Response**: While transfer attacks on classifiers have been extensively researched, the multi-bit nature of watermarking introduces significant differences. Our work addresses these unique challenges by proposing novel target watermark selection methods tailored to surrogate models and designing specialized ensemble strategies for aggregating these models. These contributions distinguish our approach from prior works on classifier transfer attacks.
> > >
> > > We also complement our empirical findings with a rigorous theoretical analysis of transferability in the context of watermarking. This analysis is rooted in the inherent characteristics of watermarking and offers insights that are not directly applicable to classifier-based transfer attacks.
> > >
> > > We are encouraged by your intent to score our work as 7 and your positive inclination toward acceptance. Thank you again for recognizing the contributions and merits of our work.

---

### Official Review · Reviewer_DtqA · 2024-11-04

**Soundness:** 3
**Presentation:** 2
**Contribution:** 2
**Rating:** 6
**Confidence:** 4

**Summary:**

The paper proposes a transfer attack to (learned/learning-based) image watermarking that first trains a lot of surrogate watermarking models (encoders + decoders), then uses PGD to find a perturbation to the watermarked images that would change the extracted watermarks by the surrogate watermarking models.

It is argued that such approach requires smaller perturbations to break learning-based watermarking when being compared to common post-processing such as JPEG, Gaussian noise, Gaussian blur, and Brightness/Contrast; and it is more effective than existing transfer attacks.

**Strengths:**

The robustness of image watermarking in no-box setting (i.e. with no access to watermarking models & with no access to detection API) is indeed a relatively overlooked area. Given that there are empirical ways to at least restrict the access to models/APIs, I consider this angle meaningful and worth investigating.

The presentation is clear enough overall and originality of this paper is not an issue.

**Weaknesses:**

My primary concerns regarding this paper are:
1. **No analysis of the computational cost in comparison with existing transfer attacks, which should be discussed clearly**. I think while the proposed method definitely incur a much larger overhead compared to previous attacks, it could still be ok to highlight the risks, providing that this limitation is presented upfront in the paper. Some analysis + some empirical numbers of computation time (with the corresponding compute resources) could be helpful for illustration.

2. **The comparison of the proposed attack with existing transfer attacks is somewhat inconclusive.**
This is because, as the authors must already know, there is a trade-off between evasion rate and the visibility/budget of the attacks (perturbation in L2/perturbation in Linf/SSIM/...). Thus a good comparison should at least compare the evasion rates with **all** baselines in representative budgets (if the other baselines work in such budgets). However, the comparison in the paper claims the proposed method is better than all others because it beats some baselines when r<= 0.1, and it beats MI-CWA and DiffPure assuming a SSIM>0.9. This is a rather misleading way of comparison, as some attacks will work better under certain (metrics of) budgets but not the others.

3. **The effectiveness of the proposed method as a "no-box" attack remain unclear, specifically regarding the transferability to different watermarking methods.**
While there are experiments involving a different target watermarking method from the surrogate ones, it is worth noting that these methods are somewhat similar (or as the authors called, they are learning-based watermarking). It is claimed that the proposed method also work for non-learning based methods (figure 13), but details regarding this set of experiments, especially what are the target watermarking methods, are missing. To my understanding, there is no rule preventing the victim using a watermarking scheme that is very different or even designed to be different from the ones used by the attacker as surrogates. This is not a deal breaker but such discussion seems to be missing.

**Questions:**

Please see Weaknesses for my primary concerns, which also include some of my suggestions/questions for the authors.

---

> ### Author Response · Authors · 2024-11-20
> **Response to Reviewer DtqA**
>
> We sincerely appreciate your constructive comments and insightful suggestions. Please find our responses below.
>
> > **Weakness 1**: No analysis of the computational cost in comparison with existing transfer attacks, which should be discussed clearly.
>
> **Response**: We appreciate your suggestion to analyze the computational cost of our transfer attack. To address this, we include the following table, summarizing the training and inference times for our attack alongside six baseline transfer attacks evaluated in our experiments.
>
> | **Method**               | **Training Time (h)**            | **Inference Time per Image (s)** |
> |--------------------------|-----------------------------------|-----------------------------------|
> | AdvEmb-RN18              | $\Delta_{\text{RN18}}$           | 72.09                            |
> | AdvCls-Real&WM           | $\Delta_{\text{RN18}}$ + 0.11    | 77.04                            |
> | AdvCls-Enc-WM1&WM2       | $\Delta_{\text{RN18}}$ + 0.11    | 77.70                            |
> | MI-CWA                   | $\Delta_{\text{RN18}}$ + 0.11    | 27.01                            |
> | DiffPure                 | $\Delta_{\text{Diff}}$           | 8.96                             |
> | WEvade-B-S               | 17.06                            | 187.03                           |
> | Ours                     | 6.50                             | 177.97                           |
>
> Training time refers to the duration required to train a surrogate model, while inference time denotes the time needed to optimize the perturbation for an image. All measurements were conducted on a single NVIDIA RTX-6000 GPU. Specifically, $\Delta_{\text{RN18}}$ represents the pre-training time of ResNet-18 on ImageNet, and $\Delta_{\text{Diff}}$ refers to the training time of the unconditional diffusion model used by DiffPure. The target watermarking model is a ResNet, and all perturbations are optimized under the constraint $SSIM \geq 0.9$. The evasion rates corresponding to each method are provided in Figure 11 in the Appendix of our revised paper.
>
> For our method, the training time is 6.5 hours. Although extending this to multiple surrogate models increases the overall duration, these models can be trained in parallel across GPUs and reused to attack  different target watermarking models, which keeps the overhead manageable. While our inference time is longer than that of five out of six baselines, it remains acceptable for attackers with adequate computational resources. A discussion of this computational cost analysis has been added to Section 7 of our revised paper.
>
> > **Weakness 2**: The comparison of the proposed attack with existing transfer attacks is somewhat inconclusive.
>
> **Response**: Thank you for highlighting this. As you suggested, we conducted an experiment comparing all attacks across different  perturbation constraints. The below table and Figure 11 in the Appendix of our revised paper presents the evasion rates of all attacks across different $SSIM$ constraints when the target watermarking model is ResNet. Our results show that our transfer attack consistently outperforms all baseline attacks.
> | **Method**               | **1.00** | **0.99** | **0.95** | **0.90** | **0.85** | **0.80** | **0.75** | **0.70** | **0.65** | **0.60** |
> |---------------------------|----------|----------|----------|----------|----------|----------|----------|----------|----------|----------|
> | AdvEmb-RN18              | 0.00     | 0.00     | 0.00     | 0.00     | 0.00     | 0.01     | 0.01     | 0.02     | 0.06     | 0.12     |
> | AdvCls-Real&WM           | 0.00     | 0.00     | 0.00     | 0.00     | 0.00     | 0.00     | 0.00     | 0.00     | 0.00     | 0.00     |
> | AdvCls-Enc-WM1&WM2       | 0.00     | 0.04     | 0.10     | 0.10     | 0.10     | 0.10     | 0.10     | 0.10     | 0.10     | 0.10     |
> | MI-CWA                   | 0.00     | 0.00     | 0.00     | 0.03     | 0.07     | 0.12     | 0.20     | 0.34     | 0.64     | 0.71     |
> | DiffPure                 | 0.00     | 0.00     | 0.00     | 0.04     | 0.19     | 0.33     | 0.73     | 1.00     | 1.00     | 1.00     |
> | WEvade-B-S               | 0.00     | 0.002    | 0.002    | 0.002    | 0.002    | 0.002    | 0.002    | 0.002    | 0.002    | 0.002    |
> | Ours                     | 0.00     | 0.218    | 0.997    | 1.00     | 1.00     | 1.00     | 1.00     | 1.00     | 1.00     | 1.00     |

---

> ### Author Response · Authors · 2024-11-20
> **Response to Reviewer DtqA**
>
> > **Weakness 3**: The effectiveness of the proposed method as a "no-box" attack remains unclear, specifically regarding the transferability to different watermarking methods.
>
> **Response**: Thank you for highlighting this. Non-learning-based watermarks lack robustness even against common perturbations, as shown in the below table (comparing evasion rates of non-learning-based watermarking models and HiDDeN under Gaussian noise with varying standard deviations) and Figure 6(c) of our revised paper.
>
> | **Method**          | **0.00** | **0.04** | **0.08** | **0.12** | **0.16** | **0.20** |
> |----------------------|----------|----------|----------|----------|----------|----------|
> | Tree-Ring           | 0.00     | 0.04     | 0.19     | 0.41     | 0.53     | 0.65     |
> | RingID              | 0.00     | 0.87     | 1.00     | 1.00     | 1.00     | 1.00     |
> | Gaussian Shading    | 0.00     | 0.005    | 0.03     | 0.09     | 0.32     | 0.46     |
> | HiDDeN             | 0.00     | 0.00     | 0.00     | 0.00     | 0.01     | 0.01     |
>
> For this reason, our study primarily focuses on learning-based watermarks. Moreover,  Figure 15 shows that our attack is also effective for non-learning-based methods. Specifically, the non-learning-based watermarking method used in Figure 15 is the DWT-DCT technique deployed by Stable Diffusion. Unlike the learning-based methods examined in our experiments, DWT-DCT is a non-learning-based watermarking approach. It leverages the Discrete Wavelet Transform (DWT) to decompose an image into frequency sub-bands, applies the Discrete Cosine Transform (DCT) to blocks within selected sub-bands, and embeds the watermark by modifying specific frequency coefficients. The watermarked image is subsequently reconstructed using inverse transforms.
>
> To clarify this distinction, we have added further discussion on non-learning-based watermarks in Section 7.

---

> ### Author Response · Authors · 2024-11-22
>
> Dear Reviewer,
>
> Thank you once again for your constructive comments and valuable feedback! We have thoroughly addressed your suggestions by conducting new experiments and revising our paper. Could you please kindly review our responses and let us know if you have any further comments or suggestions?
>
> We would greatly appreciate it if you could reconsider your score in light of our new experimental results and the improvements made in the paper.
>
> Thank you for your time and consideration!

---

> > ### Comment · Reviewer_DtqA · 2024-11-22
> >
> > Thank you for the reminder & for your work in putting together the rebuttal.
> >
> > Summary: I think Weakness 1 & 2 are at least partially addressed; I am not convinced by the rebuttal targeting weakness 3 and I will explain why.
> >
> > For weakness 1: This is consistent with my previous expectations. The proposed method have indeed a much larger overhead compared to others, especially that the reported numbers are for a single surrogate model. This is obviously a limitation and in my opinion, it actually does not look better with claims such as "it remains acceptable for attackers with adequate computational resources", unless you further explain the reasoning behind such claims. I would suggest to edit the corresponding discussion (newly added to Sec 7) to be upfront about such limitations. But overall I don't think this will be a big issue anymore for the efficiency of discussions.
> >
> > For weakness 2, these results look good. Could you also provide details regarding how you apply/enforce the SSIM constraints for each baseline reported in this table? These would help the readers (and me) to better understand the implications of these numbers.
> >
> > For weakness 3: Firstly, thank you for adding the details. That being said, one of my concern remains: "To my understanding, there is no rule preventing the victim using a watermarking scheme that is very different or even designed to be different from the ones used by the attacker as surrogates." Could you please elaborate on this?
> >
> > This is not necessarily a deal breaker but I think the current presentation of the submission is misleading in a sense that 1. the corresponding discussion is missing and 2. the claims seem to be suggesting that the proposed attacks still work under such circumstances (e.g. from abstract: "Our major contribution is to show that, both theoretically and empirically, watermark-based AI-generated image detector is not robust to evasion attacks even if the attacker does not have access to the watermarking model nor the detection API").
> >
> > I would suggest the authors to explicitly tune down the claims if they do not claim effectiveness against sufficiently different/adaptive defenses, or explain the reasonings behind if they believe the proposed attack should remain effective in such cases.

---

> > > ### Author Response · Authors · 2024-11-22
> > > **Response to Reviewer DtqA**
> > >
> > > Thank you for your timely reply and constructive comments, which have been invaluable in helping us improve our paper.
> > >
> > > We are pleased to note that our newly added experiments and the revised paper partially address Weaknesses 1 and 2. Regarding Weakness 3, we sincerely appreciate your acknowledgment that it is not necessarily a deal breaker.
> > >
> > > > For weakness 1: This is consistent with my previous expectations. The proposed method have indeed a much larger overhead compared to others, especially that the reported numbers are for a single surrogate model. This is obviously a limitation and in my opinion, it actually does not look better with claims such as "it remains acceptable for attackers with adequate computational resources", unless you further explain the reasoning behind such claims. I would suggest to edit the corresponding discussion (newly added to Sec 7) to be upfront about such limitations. But overall I don't think this will be a big issue anymore for the efficiency of discussions.
> > >
> > > **Response**: Thank you for highlighting this. Based on your suggestion, we have revised the discussion in Section 7 to explicitly acknowledge this limitation.
> > >
> > > > For weakness 2, these results look good. Could you also provide details regarding how you apply/enforce the SSIM constraints for each baseline reported in this table? These would help the readers (and me) to better understand the implications of these numbers.
> > >
> > > **Response**: Thank you for your insightful questions. Since the baseline methods control their attack strength using the $\ell_{\infty}$ norm of the perturbation, we also regulate the SSIM of the perturbed images by setting an $\ell_{\infty}$ norm budget for each method. Specifically, we perform a hyperparameter search to determine the maximum $\ell_{\infty}$ norm budget for each method such that the SSIM of their perturbed images remains within our predefined SSIM constraints.
> > >
> > > > For weakness 3: Firstly, thank you for adding the details. That being said, one of my concern remains: "To my understanding, there is no rule preventing the victim using a watermarking scheme that is very different or even designed to be different from the ones used by the attacker as surrogates." Could you please elaborate on this? This is not necessarily a deal breaker but I think the current presentation of the submission is misleading in a sense that 1. the corresponding discussion is missing and 2. the claims seem to be suggesting that the proposed attacks still work under such circumstances (e.g. from abstract: "Our major contribution is to show that, both theoretically and empirically, watermark-based AI-generated image detector is not robust to evasion attacks even if the attacker does not have access to the watermarking model nor the detection API"). I would suggest the authors to explicitly tune down the claims if they do not claim effectiveness against sufficiently different/adaptive defenses, or explain the reasonings behind if they believe the proposed attack should remain effective in such cases.
> > >
> > > **Response**: Thank you for highlighting this and noting that it is not necessarily a deal breaker. Our evaluation is about existing watermarking methods. However, we cannot prevent the victim from designing a completely novel and private watermarking method. In such a scenario, the effectiveness of our transfer attack is unclear. It is an interesting future work to explore such watermarking methods and evaluate the effectiveness of our transfer attack.
> > >
> > > We have added a discussion on this limitation to Section 7 of our revised paper. Moreover, to explicitly tune down our claims, we have revised the abstract and introduction to emphasize that our empirical analysis demonstrates the effectiveness of our attack on existing watermarking methods but the transferability to entirely different and unknown watermarking methods is unclear.

---

> > > ### Author Response · Authors · 2024-11-25
> > >
> > > Dear Reviewer,
> > >
> > > We wanted to kindly remind you about the responses and updates we shared regarding your constructive comments and valuable feedback on our paper. We have addressed your suggestions by conducting new experiments and revising the manuscript accordingly.
> > >
> > > Could you please review our responses at your earliest convenience? If you have any further comments or suggestions, we would be more than happy to address them. Additionally, we kindly ask you to reconsider your score in light of the new experimental results and the improvements made in the paper.
> > >
> > > As the discussion phase deadline is approaching, we greatly appreciate your time and attention to this matter.
> > >
> > > Thank you again for your thoughtful feedback and for considering our revised work.

---

> > > > ### Comment · Reviewer_DtqA · 2024-11-25
> > > >
> > > > Thanks again for the reminder.
> > > >
> > > > Based on your efforts made during discussion, I am now relatively comfortable with voting positively for this submission. Thus I am gonna raise my score to marginally above the acceptance threshold.
> > > >
> > > > You don't need to finish this during the discussion period, but regarding weakness 2, **"Since the baseline methods control their attack strength using the $\ell_{\infty}$ norm of the perturbation, we also regulate the SSIM of the perturbed images by setting an $\ell_{\infty}$ norm budget for each method."** seems rather unfair to the baseline methods as they are optimized for $\ell_{\infty}$ norm.
> > > >
> > > > Since your method also operate on the $\ell_{\infty}$ norm (eq (3)), why reporting only SSIM? This part does not look very good, but since my reply delayed, I am not factoring this in my scoring. I would still like the authors to include comparisons under $\ell_{\infty}$ norm for later versions.
> > > >
> > > > Have a good day!

---

> > > > > ### Author Response · Authors · 2024-11-25
> > > > > **Response to Reviewer DtqA**
> > > > >
> > > > > Thank you for your constructive comments and for raising the score. Based on your suggestion, we will begin running experiments to compare all baseline methods under the same $\ell_{\infty}$ norm constraint. While we may not be able to complete these experiments during the discussion period, we will include the results in the final version of the paper.
> > > > >
> > > > > We sincerely appreciate your positive feedback and your support in voting positively for our paper.

---

### Meta-Review · Area_Chair_Ampk · 2024-12-21

**Metareview:**

2x accept, 2x borderline accept. This paper introduces a no-box scenario attack framework that trains multiple surrogate watermarking models and ensembles them to generate perturbations that remove or alter watermarks in AI-generated images without access to the target model. The reviewers agree on the (1) strong empirical performance on diverse watermarking methods, (2) comprehensive experiments and theoretical discussions, and (3) clear, well-structured presentation of the attack pipeline. However, they note (1) considerable computational overhead, (2) limited coverage of completely different watermark schemes (especially non-learning-based), and (3) partial reliance on theoretical assumptions requiring further clarifications. The authors have followed up with detailed responses and additional experiments addressing these concerns—showing acceptable runtime through parallelization, expanding coverage of non-learning-based methods, and refining theoretical justifications—so the AC leans to accept this submission.

**Additional Comments On Reviewer Discussion:**

N/A

---

### Decision · Program_Chairs · 2025-01-22

Accept (Poster)